# Programming viscoelastic properties in a complexation gel composite by utilizing entropy-driven topologically frustrated dynamical state

Gui Kang Wang [1,2], Yi Ming Yang[1,2] & Di Jia [1,2] ✉

Hydrogel composites in an aqueous media with viscoelastic properties and elastic modulus that can be precisely tailored are desirable to mimic many biological tissues ranging from mucus, vitreous humor, and nucleus pulposus as well as build up biosensors. Without altering the chemistry, tuning the physical interactions and structures to govern the viscoelastic properties of the hydrogels is indispensable for their applications but quite limited. Here we design a complexation gel composite and utilize the physical principle of topologically frustrated dynamical state to tune the correlated structures between the guest polycation chains and negatively charged host gels. We precisely quantify the mesh size of the host gel and guest chain size. By designing various topologically correlated structures, a viscoelastic moduli map can be built up, ranging from tough to ultrasoft, and from elastic-like with low damping properties to viscous-like with high damping properties. We also tune the swelling ratio by using entropy effect and discover an Entropy-driven Topologically Isovolumetric Point. Our findings provide essential physics to understand the relationship between entropy-driven correlated structures and their viscoelastic properties of the complexation hydrogel composites and will have diverse applications in tissue engineering and soft biomaterials.

Hydrogels with moduli in the range of $1-10^5$ Pa are of widespread interest as biomaterials. Many native tissues have moduli in this range, including vitreous humor, nucleus pulposus, cartilage, kidney cortex, adventitial layer, etc. One ultrasoft example is the extracellular matrix of articular cartilage, which is comprised of two structurally different components, including large aggregating proteoglycan and type II collagen. Although the mechanical properties of cartilage can be greatly influenced by the related structures and components of collagen and proteoglycans, it is still unclear how they can precisely provide the mechanical properties for cartilage materials. For the in vitro mixture of proteoglycan and collagen, the shear modulus is always less than 100 Pa and is smaller than the loss modulus, which indicates they are predominantly viscous in nature[1]. Another soft case is the vitreous humor, which fills the posterior cavity of the eye. Such a native tissue in vitro has elastic modulus in the order of 50–120 Pa[2–4].

And for the medium soft case is the nucleus pulposus, which has a frequency-sensitive viscoelastic behavior[5], has an elastic modulus G′ in the range of 3–7 kPa tested over the range of angular frequencies 1–100 rad/s. The mechanical damping properties are also required for soft biological tissues, and a viscosity component is required for efficient damping. Chemical gels with covalent bonds have stable mechanical properties, but it is challenging to change their moduli without introducing specific new chemistry. In contrast, the mechanical properties of physical gels are more tunable by changing the

[1]Beijing National Laboratory for Molecular Sciences, Laboratory of Polymer Physics and Chemistry, Institute of Chemistry, Chinese Academy of Sciences, Beijing 100190, China. [2]University of Chinese Academy of Sciences, Beijing 100049, China. ✉e-mail: jiadi11@iccas.ac.cn

conditions of temperature, pH, ionic strength, etc. In this study, we have designed a complexation composite with the negatively charged gel matrix as the host and polycations as guest chains. In addition to the traditional methods to tune the mechanical properties such as tuning the temperature, pH, ionic strength, etc.[6,7], here we have designed a new strategy to precisely tailor the viscoelastic properties of the gel composites by utilizing the topological correlated structures between the guest and host arising from long-chain connectivity and large conformational entropy. A comparison of the complexation gel composites in this work with synthetic hydrogels and soft tissues has been made. It clearly shows that our complexation gel composites mainly fill the gap in the ultrasoft-soft regime (Fig. 1).

Traditionally, polymers under confinements are all diffusive and obey Einstein's diffusion law, no matter they are under weak confinement like the Ogston model, single entropic barrier model, or under very strong confinement like reptation model, in which the polymer chains can still do one-dimensional random walk and move forward like a snake[8–14]. The above mechanisms of polymers being diffusive under confinement have been verified by experiments in the past several decades[15–23]. Recently, we have discovered a new dynamical state defined as non-diffusive topologically frustrated dynamical state (TFDS), in which the polymer chains in the gel matrix do not diffuse defying Einstein's diffusion law, due to the emergence of extreme metastability from freezing of chain's conformational entropy at intermediate confinement[24–26]. The condition of intermediate confinement is when the polymer chain size $R_g$ is larger than the gel mesh size $\xi$, and $\xi$ is much larger than the polymer segmental length $l$, that is $R_g > \xi \gg l$. At intermediate confinements, the polymer chains can reside in several meshes of the gel network, so that the polymer chains become non-diffusive topologically frustrated dynamical state (TFDS) due to the multiple entropic barriers. Therefore, in the TFDS regime, the center of mass of the chains are non-diffusive and in a long-lived metastable state, but within one polymer chain, the polymer segments are still able to have hierarchically segmental dynamics such as Zimm or Rouse dynamics. While the polymer chains can diffuse outside the TFDS regime.

Tuning the topological correlated structures between the guest chains and host gel network and making the system cross the boundary of TFDS can not only facilitate the release of macromolecular cargos using hydrogel-like host matrices as well as build molecular machines that depend on memory associated with metastability but also offer a diverse set of opportunities to controllably tailor the viscoelastic properties of the complexation composites. As schematically illustrated in Fig. 1a–d, in the complexation composite with the negatively charged gel matrix as the host and polycations as the guest chains, by tuning the guest chain length with fixed host gel mesh size, we can precisely control whether they are inside or outside the TFDS regime. For short guest chains whose chain size is smaller than the mesh size and are outside the TFDS regime, they can complex with the gel strands within each individual gel mesh. While for very long guest chains whose chain size is much larger than the mesh size, such that one chain can cover more than 40 gel meshes calculated in three-dimensional space and they are inside the TFDS regime, in this case, one guest chain is able to cooperatively complex with the gel strands from multiple gel meshes. Such an entropy-driven TFDS can make a topologically quasi-interpenetrating network and change their viscoelastic properties significantly. Another key tuning parameter, which can change their viscoelastic properties, is the molar charge ratio of the guest polycations over the negatively charged host gels. By tuning the guest polycation concentrations in the fixed host gel matrix, the charges on the gel matrix can be either partially complexed or overcomplexed by the polycation guest chains. This will not only change the number of physically ionic bonds in the gel composites but also change the topologically correlated structure

between the guest and host of the gel composites, thus leading to different viscoelastic properties.

Besides, we have also found that in the chemical reaction, the polycation guest chains mixed with the negatively charged monomers in the pre-gel solutions can influence the charged monomer chemical conversion and thus the number of chemical bonds in the gel composites. Because the negatively charged monomers repel each other and cannot come close to form the chemical bonds with each other easily due to the electrostatic repulsion, the addition of guest polycations can neutralize the charges on the negatively charged monomers and facilitate the formation of chemical bonds and increase their monomer chemical conversion. However, when the guest polycation concentration is too high, the excess guest chains will act as defects and the steric hindrance effect will dominate during the chemical reaction, leading to the decrease of the chemical conversion. Therefore, such a physical complexation effect provides a new strategy to in turn control the chemical conversion rate in the charged gel composites.

Overall, the viscoelastic properties of the guest-host complexation gel composite system can be quantitatively tuned in a broad range, ranging from tough gel to ultrasoft gel, and from elastic-like gel with low damping properties to viscous-like gel with high damping properties (Fig. 1e, f). Tanδ values of the gel complexation composites can be tuned in the range of 0.06–1, which can match various soft tissues. For example, for bovine meniscus, tanδ is in the range of 0.19–0.38 for different parts[27], and for bovine articular cartilage, tanδ is around 0.13–0.2[1,28]. For human lumbar, tan is in the range of 0.42–0.58[5]. Besides, Larson et al. studied a broad range of polymer concentration and salt concentration in polyelectrolyte coacervates[29]. Here we have studied the different charge ratio between the guest polyelectrolytes and host gels, and also the molecular weight dependence of the guest chains in the complexation gel composites.

In addition to the broad range of viscoelastic properties, we have also discovered an Entropy-driven Topologically Isovolumetric Point for the complexation gel composite systems at the swelling equilibrium. When the gel has reached its swelling equilibrium, the total osmotic pressure becomes zero, and several physical factors contribute to the osmotic pressure, such as Donnan equilibrium, free energy due to mixing, concentration fluctuations, gel elasticity, and electrostatic correlations between polymer segments, etc. Due to the Donnan equilibrium, the mobile counterions exert osmotic pressure on the polyelectrolyte gels and make the polyelectrolyte gel swell to a much larger extent in water at low ionic strength compared to the uncharged gels[30–34]. External stimuli such as solvent quality, temperature, pH, and salt concentrations can influence the swelling equilibrium of the gels. For example, McCoy et al. found that a negatively charged gel in the acetone/water mixture solvent can experience a discontinuous volume collapse with the increase of NaCl salt concentration[35]. Other studies have shown that polyampholyte gels, which contain both positive and negative charges in the gel strands, swell at asymmetric charge ratio but shrink at stoichiometric charge ratio due to the formation of microphase separation domains inside the gels[36–38].

In this work, we have designed a new strategy to tune the swelling ratio of the complexation composite by only changing their topologically correlated structures between the guest and host. By tuning the size ratio of the guest polycation chains $R_g$ over the mesh size of the host gel $\xi$, the gel composite will realize a repulsive gel composite to semi-interpenetrating network to topologically quasi-interpenetrating network structural transition and they will change from swollen state to shrunk state at swelling equilibrium as $R_g$ increases at fixed $\xi$. And we have found an isovolumetric point where the gel composite volume neither shrinks nor swells. Such an isovolumetric point is entropy-driven by the topologically correlated structures between the guest chains and host gels. Therefore, we have defined it as Entropy-driven

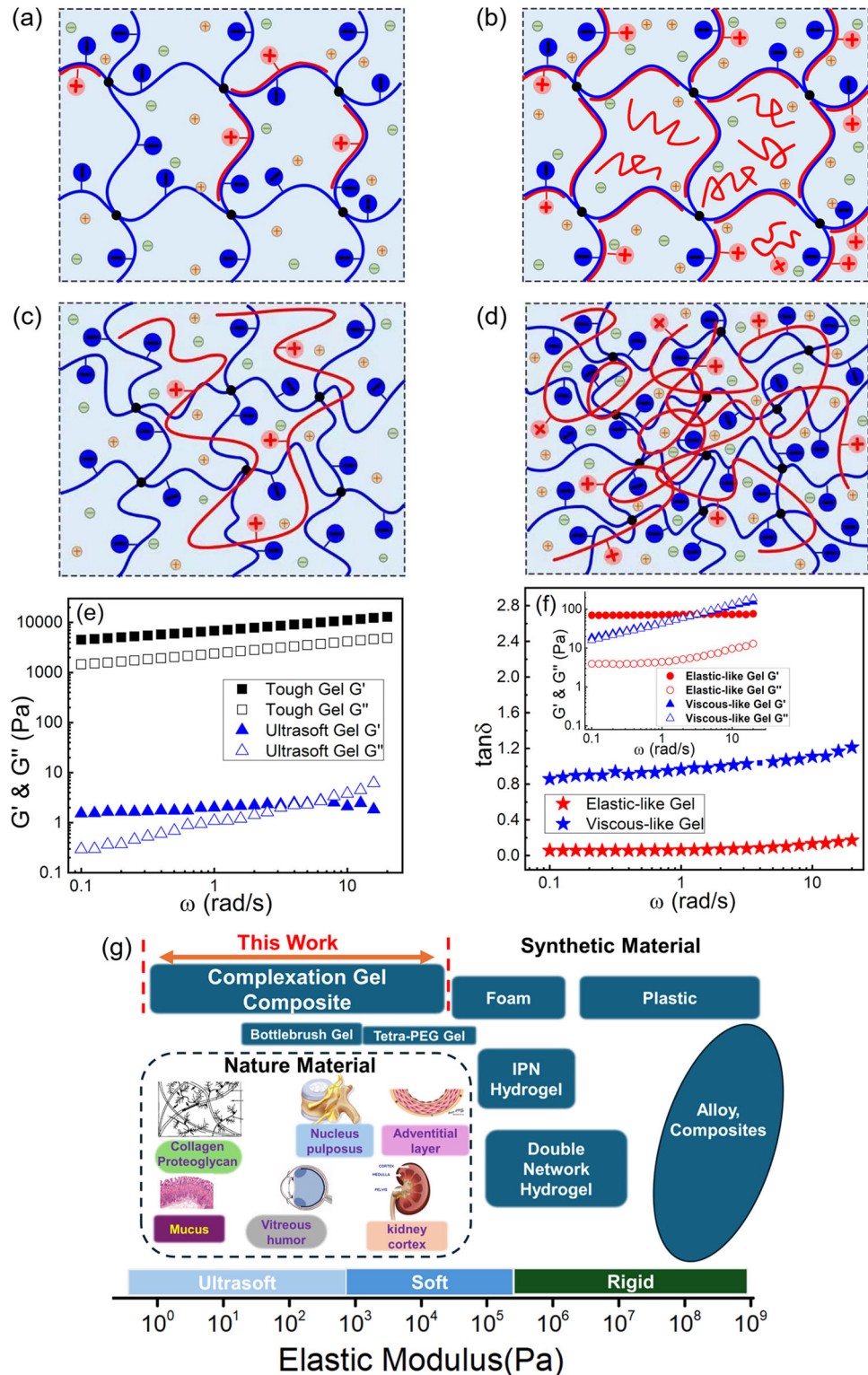

**Fig. 1 | Complexation gel network structures and their viscoelastic properties.**
**a**–**d** Schematic illustration of the complexation structure between short (**a**,**b**) and long (**c**,**d**) guest polycations and negatively charged host gel. When the charges on the gel are not fully complexed (**a**, **c**) and fully or overcomplexed (**b**, **d**) by guest polycations. **e**, **f** Experimental results for the guest-host complexation system, exhibiting a broad range of viscoelastic properties, from tough gel to ultrasoft gel (**e**), and from elastic-like gel with low tanδ to viscous-like gel with high tanδ (**f**). Inset is their dynamic frequency sweeps. **g** A comparison of elastic modulus for both natural soft tissues, synthetic materials, and the complexation gel composites in this work[1–5,28,71–76]. Reproduced with permission from ref. [28]. Copyright 2005, John Wiley & Sons.

Topologically Isovolumetric Point for the complexation gel composite systems at swelling equilibrium. Tuning the swelling ratio solely by utilizing conformational entropy and topologically correlated structures may provide a new strategy to control the swelling ratio of charged gel composites.

## Results

### Introduction of the host gel-guest chain complexation system

The gel composite is a complexation of the negatively charged host gel and gust polycation chains. Specifically, the negatively charged host gel matrix is poly (acrylamide-co-sodium acrylate) (PAm-PAc) gel with 80% charge density and 1.5% crosslinking density, and the guest polycation is poly (diallyldimethylammonium chloride) (PDA). The gel composites were synthesized with PDA guest chains in the pre-gel solutions and then initiated the free radical polymerization so that PDA guest chains can be uniformly distributed in the host gel. Please note that no chemical bonds can be formed between PDA guest chains and the PAm-PAc host gels so that only physical interactions exist between the guest chains and the host gels.

In this study, there are three key experimental handles to tune the viscoelastic properties: the size ratio of the guest PDA chain size $R_g$ over the mesh size $\xi$ of the gel matrix, the apparent molar charge ratio $r$ in feed in the pre-gel solutions (positive charged monomers of PDA/negatively charged monomers of the gel matrix), and the swelling methods. Please note that although element analysis method was used to determine the true anion charge fraction in polyampholyte gels[36,39], our system is more complicated so that it is hard to determine the true molar charge ratio by using element analysis. Specifically, the mesh size of the gel matrix is fixed, and three molecular weights of PDA guest chains ($M_w < 100$ kDa named as SPDA; $M_w = 200-35$ kDa named as MPDA; $M_w = 400-500$ kDa named as LPDA) were used to tune the size ratio $R_g/\xi$. We have chosen the molar charge ratio $r = 0, 0.3, 0.6, 1,$ and 2, respectively. The molar charge ratio $r$ was tuned by changing PDA guest chain concentrations at the fixed host gel matrix. Besides, after the gel composites were synthesized, three states were studied by rheology: as-prepared samples without swelling, samples swelled by deionized water, and samples swelled by PDA solutions, whose PDA concentration is the same as that inside the gel composites. The samples were swelled for more than one week until their volumes no longer changed anymore, indicating they reached swelling equilibrium. All the experiments were carried out at room temperature.

### Quantification of the mesh size and elasticity for the host gel

In order to characterize the guest polycation chains and charged host gels by dynamic and static light scattering, 0.5 M NaCl was added to screen the electrostatic interactions. The mesh size $\xi$ of the PAm-PAc gel network is obtained from the static light scattering by using the Ornstein-Zernike equation[24,32]:

$$I(q) = \frac{I(q \to 0)}{1 + q^2\xi^2} \tag{1}$$

where $q = (4\pi n/\lambda)\sin(\theta/2)$ is the scattering wavevector with the scattering angle $\theta$, wavelength of the incident light $\lambda$, and refractive index of the medium $n$. $I(q)$ is the scattered light intensity at the scattering wavevector $q$, $\xi$ is the correlation length and is used here to estimate the averaged mesh size of the gel network. As shown in Fig. 2a, a plot of $1/I(q)$ versus $q^2$ is linear and we can obtain the averaged mesh size of the host gel $\xi_{static} = (21.0 \pm 2.0)$ nm by static light scattering.

For gels, dynamic light scattering can quantify the elasticity of the gels. The electric field correlation function $g_1(q, t)$ is proportional to the correlation of the gel strands displacement along the longitudinal direction $u_l$, expressed as follows[40,41]:

$$g_1(q,t) \sim \langle u_l(q,0)u_l(q,t)\rangle = \langle u_l(q,0)^2\rangle e^{-\Gamma t} \tag{2}$$

where $t$ is the correlation time. $\Gamma = D_{gel}q^2$ is the relaxation rate, in which $D_{gel}$ is the elastic diffusion coefficient of the gels $D_{gel} = M/f$ (M is the longitudinal modulus of the gel, and $f$ is the friction coefficient between the gel strands and the solvent). $D_{gel}$ is proportional to the longitudinal modulus M of the gels, so that $D_{gel}$ indicates the gel elasticity.

In general, there are several relaxation modes that can contribute to gel dynamics, but we can only probe the slower modes with dynamic light scattering[40,42–47]. The electric field correlation function $g_1(q, t)$ of the gel matrix at scattering angle 30° is shown in Fig. 2b as a representative, it can be fitted by three exponential decays, the quality of the fitting can be seen from the fitting residuals in Fig. 2b. And the corresponding relaxation time distribution function $f(t)$ at scattering angle 30° can be obtained from CONTIN fitting method (Fig. 2c)[48]. There are three relaxation modes at each angle, but only the second relaxation mode is dominant and diffusive, while the first and third modes are non-diffusive, due to the inherent inhomogeneities from nonergodicity of the gel network. If one relaxation mode is diffusive, the relaxation rate $\Gamma$ at different scattering angles have the quadratic dependence on the scattering wavevector $q$, and from the slope of relaxation rate $\Gamma/q^2$ we can obtained the elastic diffusion coefficient of the gels $D_{gel} = (1.25 \pm 0.05)E-7$ cm²/s (Fig. 2d), and the relaxation rate $\Gamma$ of other two relaxation modes do not have quadratic q dependence expected for diffusive modes (Fig. 2e, f). Besides, the gel diffusion coefficient $D_{gel}$ can also be interpreted as the cooperative diffusion coefficient related to the dynamic correlation length of concentration fluctuations of the gel, analogous to that in the semidilute solutions[49,50]. So $D_{gel}$ can be converted to the dynamic correlation length $\xi_{dynamic}$ by $\xi_{dynamic} = k_BT/6\pi\eta D_{gel}$ (with $k_B$, T, and $\eta$ are the Boltzmann constant, the absolute temperature, and the solvent viscosity, respectively), thus we obtain the dynamic correlation length $\xi_{dynamic} = (20.6 \pm 1.6)$ nm. Please note that although the measured values of the dynamic correlation length and the mesh size obtained from static light scattering are very close, the two concepts are not equal in terms of their physics.

### Quantification of the size and shape for the guest chains

We have investigated PDA chains with three molecular weight $M_w = 400-500$ kDa, 200–350 kDa, and <100 kDa, and they are named as LPDA, MPDA, SPDA, respectively. The chain conformation and size of these polycations are characterized by measuring their radius of gyration $R_g$, and the shape factor $R_g/R_h$ ($R_h$ is the hydrodynamic radius) in 0.5 M NaCl solutions. The averaged values of $R_g$ are 47.2 nm, 38.3 nm, 15.8 nm for LPDA, MPDA, and SPDA, respectively and the corresponding shape factors of $R_g/R_h$ are 1.55, 1.58, 1.60 for LPDA, MPDA, and SPDA, indicating the chains adopt random coil conformation in solutions. We have also estimated the overlap concentrations $C^* = 1.7$ g/L, 1.9 g/L, and 10 g/L for LPDA, MPDA, and SPDA, respectively in solutions. The polydispersity $M_w/M_n$ values estimated from the cumulant analysis method from DLS data are 2.4 for LPDA, 2.2 for MPDA, and 2.3 for SPDA, respectively[51–53]. Besides, when the guest PDA guest chains are inside the host gel matrix with the averaged mesh size 21 nm, if we calculate in three-dimensional space one guest chain can cover 47 meshes, 25 meshes, and <2 meshes, respectively for LPDA, MPDA, and SPDA in the gel matrix. Here the detailed characterization for LPDA by static and dynamic light scattering is shown as a representation, the detailed characterization of MPDA and SPDA is in the Supplementary Fig 1.

For LPDA, the radius of gyration $R_g$ in dilute solution with 0.5 M NaCl was measured by using static light scattering and Guinier plot

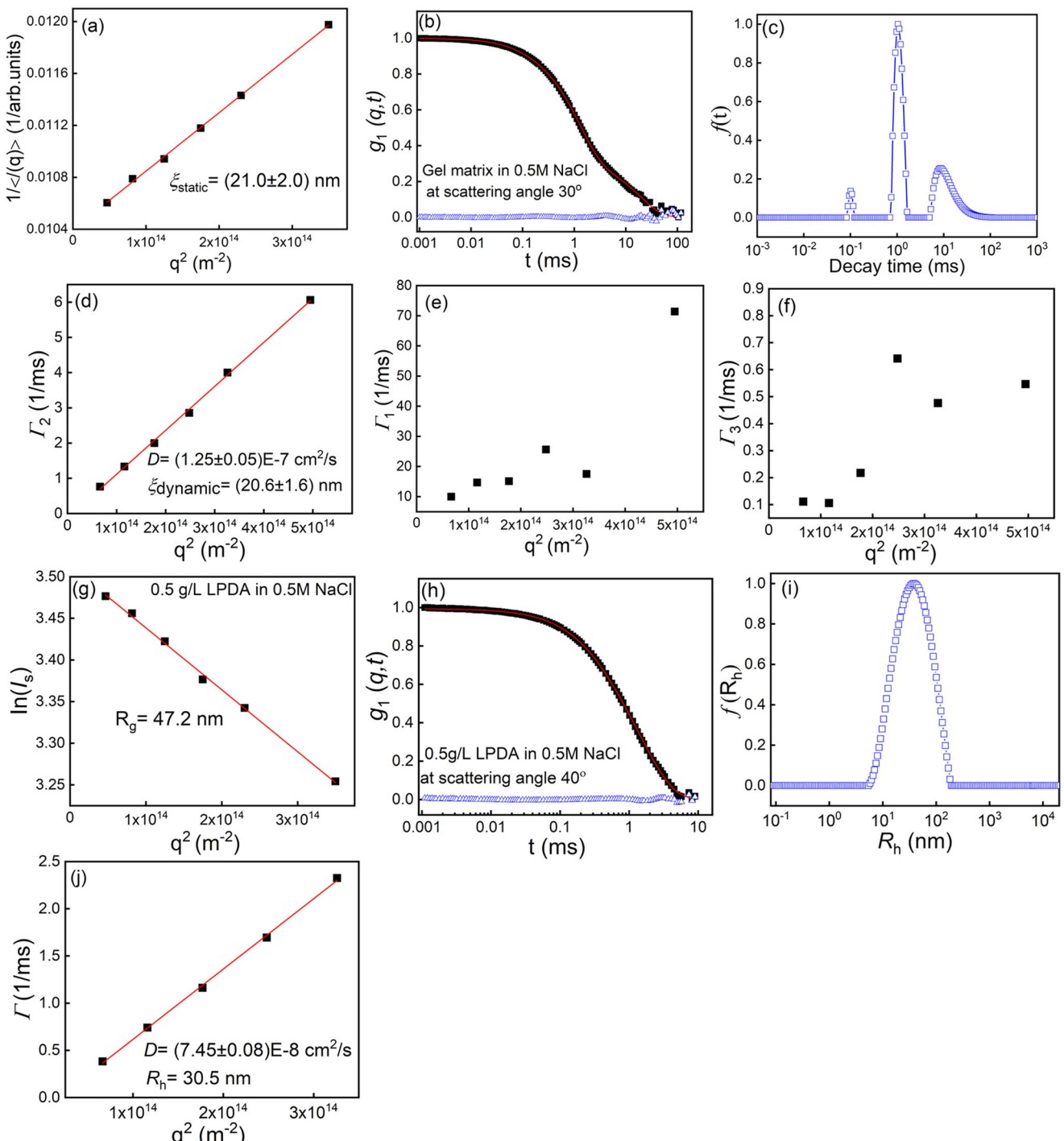

**Fig. 2 | Dynamic and static light scattering characterization for the gel matrix and PDA chains. a**–**f** Characterization of the host gel matrix with 0.5 M NaCl. **a** Ornstein-Zernike plot of the inverse average scattering intensity $1/I(q)$ versus $q^2$ for the gel matrix, $I(q)$ is averaged over 3 independent measurements. **b** Normalized field correlation function $g_1(q, t)$ at scattering angle 30° measured by dynamic light scattering. The red curve is the fitting curve fitted by three exponential decays and the blue open triangles are the residuals between the fitting curve and the raw data. **c** Corresponding relaxation time distribution function obtained from CONTIN fit at scattering angle 30° for the gel matrix. **d**–**f** $q^2$ dependence of the relaxation rate Γ of the second mode (**d**), first mode (**e**), and third mode (**f**) for the gel matrix. Only the second mode has a linear relationship with $q^2$. **g**–**j** Characterization of 0.5 g/L LPDA in 0.5 M NaCl solution. **g** Guinier plot of $\ln(I_s)$-$q^2$. **h** Normalized electric field correlation function $g_1(q, t)$ at scattering angle 40°. **i** Corresponding relaxation time distribution function at scattering angle 40°. **j** $q^2$ dependence of the relaxation rate Γ.

$I_s(q) = I_s(0) \exp(-q^2 R_g^2/3)$[54], from the slope of $\ln(I_s)$-$q^2$, we can get the averaged $R_g = 47.2$ nm (Fig. 2g). The hydrodynamic radius $R_h$ was measured by dynamic light scattering. The electric field correlation function $g_1(q, t)$ for LPDA in dilute solution with 0.5 M NaCl can be fitted well by one single exponential decay (Fig. 2h). The distribution function of the hydrodynamic radius $R_h$ is shown in Fig. 2i. From the slope of $\Gamma/q^2$ we got the diffusion coefficient $D = (7.45 \pm 0.08)$E-8 cm²/s (Fig. 2j) and was converted to hydrodynamic radius $R_h = 30.5$ nm based on the Stokes-Einstein equation[54].

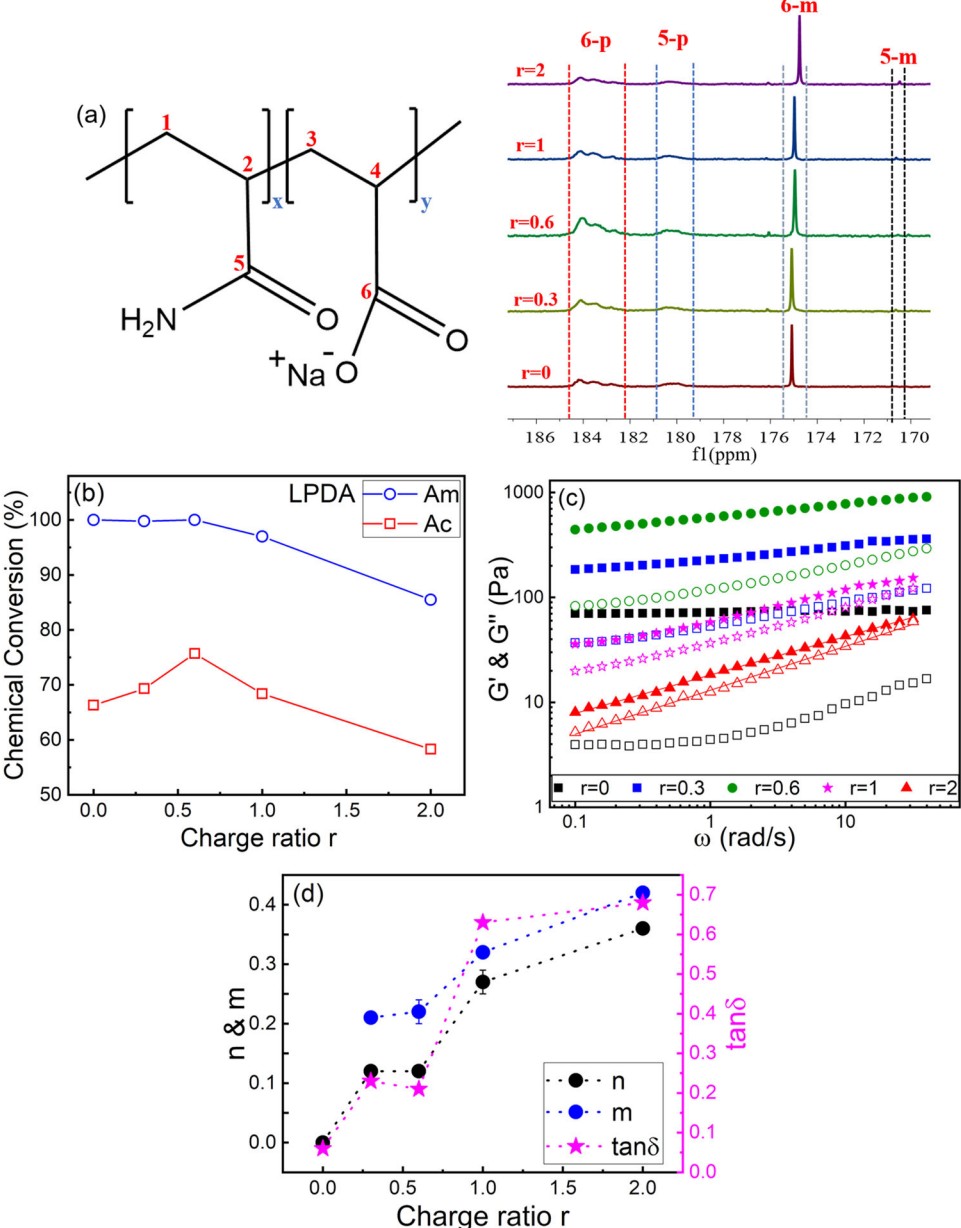

**Fig. 3 | Characterization of the as-prepared gel composite with LPDA inside at different molar charge ratio r. a** $^{13}$C NMR spectra and the chemical structure of the PAm-PAc gel composites. 5-p and 6-p indicate the peaks for polymers, and 5-m and 6-m indicate the peaks for monomers. **b** Chemical conversion of Am monomers and Ac monomers plotted against $r$. **c** Dynamic frequency sweep of gel composites. Storage moduli G' (closed symbols) and loss moduli G" (open symbols) versus sweep frequency ω at shear strain γ = 1%. The data for $r$ = 2 is shifted by /2.5 for clear vision. Red lines indicate the best fit for $r$ = 2 as a representative. **d** $n$, $m$, and tanδ plotted against $r$. G' and G" have power-law relations of G'-ωⁿ, G"-ωᵐ, and tanδ = G"/ G'. Each of the data points denotes the average of three individual replicates, and error bars are ± sd.

## Impact of the guest chains on the gelation reaction of the host gel and their viscoelastic properties

The monomer conversion of the gel composite was analyzed using $^{13}$C NMR spectra. In Figs. 3a, 5-p and 6-p indicate the peaks for PAm and PAc on the polymerized gel, while 5-m and 6-m indicate the peaks for unreacted Am and Ac monomers. From the peak areas we can get the chemical conversion of Am and Ac respectively. Figure 3b shows the chemical conversion of Am and Ac monomers as a function of molar charge ratio r for LPDA. The chemical conversion for samples with MPDA and SPDA are shown in Supplementary Fig 2.

Please note that the molar charge ratio $r$ is the number of the charges of PDA/negatively charged monomers of the host gel in the pre-gel solutions. So, with the fixed charged monomers for the host gel in the pre-gel solutions, increasing $r$ is equal to increasing PDA concentrations. Chemical conversion of neutral Am monomers decreases slightly with $r$ because of the steric hindrance arising from unreacted LPDA guest chains during the gelation reaction. However, the chemical conversion of charged Ac monomers first increases and then decreases with $r$. Because charged Ac monomers repel each other due to electrostatic repulsion so that they cannot come close and connect to form chemical bonds with each other easily. While the addition of LPDA guest polycations can neutralize some of the charges on the Ac monomers so that Ac monomers can come close to each other and form chemical bonds more easily. Thus, the chemical conversion of Ac monomers first increases with $r$. When most of the Ac monomers are neutralized with high concentration of LPDA (at high $r$), the excess

LPDA guest chains will act as defects and the steric hindrance effect will dominate during the gelation reaction. Therefore, for charged Ac monomers, the chemical conversion will first increase and then decrease with $r$, and the maximum chemical conversion of Ac monomer is 76% at $r = 0.6$.

LPDA guest polycations can not only influence the chemical conversion of the host gel but also can form physically ionic bonds with the negatively charged host gels. Both the chemical bonds and the physically ionic bonds influence the viscoelastic properties of the gel composites. As shown in Fig. 3c, the elastic modulus G′ increases as r increases from 0 to 0.6, because both the number of chemical bonds and physically ionic bonds increases. As $r$ further increases, the elastic modulus G′ decreases with $r$. On one hand, the number of chemical bonds decreases due to the decreasing of chemical conversion at higher $r$. On the other hand, at higher $r$ much more guest LPDA chains are present beyond the saturation of complexation with the host gel, so that the extra uncomplexed LPDA chains and their counterions will increase the ionic strength of the system, thus weaken the physically ionic bonds between the guest and host, leading to the decrease of G′. Such salt-induced softening effect is called saloplasticity[55]. Besides, tanδ indicates the elastic or viscous nature of the material. δ varies between 0° (Hookean solid) and 90° (Newtonian fluids)[56]. Tanδ increases from 0.1 to 0.7 as r increases from 0 to 2 (Fig. 3d), indicating the gel composites can dissipate more energy and have better-damping properties at higher $r$. As r increases, the viscoelastic properties of the gel composites transfer from predominantly elastic to more viscous. A viscosity component is required for efficient damping, and their damping properties can be controlled by changing the molar charge ratio $r$.

Moreover, for chemically crosslinked gels, the storage modulus G′ is expected to be independent of frequency, and G′ is an order of magnitude greater than the loss modulus G″, indicating the elastic nature of these gels[32,56]. While for physical gels with ionic bonds, they can be well described by the sticky Rouse motion of associating polymer strands derived from ionic bonds (sticker) with $G'\sim\omega^n$, $G''\sim\omega^m$ ($n = m = 0.5$)[55,57,58]. For the gel matrix alone with no guest PDA chains ($r = 0$), G′ has no frequency dependence ($n = 0$) and G′ is an order of magnitude higher than G″, which has a minimum, indicating the nature of chemical gels. As $r$ increases, both G′ and G″ have a stronger frequency dependence and the corresponding $n$ and $m$ values increases (Fig. 3d). Here n and m values are smaller than 0.5, deviating from the sticky Rouse relaxation of ionic associations, probably because there is a distribution for the strength of the ionic bonds[39,59–61]. Besides, the value difference between G′ and G″ also becomes smaller with increasing $r$. Therefore, as r increases, the gel composites gradually transfer from the chemical gels to more like physical gels with physically ionic bonds.

## Impact of the swelling methods on the viscoelastic properties of the complexation gel composites

For the gel matrix alone without LPDA, when it was fully swelled in the deionized water, the elastic modulus G′ decreases to roughly one-fourth of G′ for the as-prepared state before swelling (Fig. 4a). For polymer gels, the elastic modulus G′ is proportional to the gel volume fraction Φ following $G'\sim\Phi^{1/3}$ scaling law[30,62], and the gel volume fraction is defined as:

$$\Phi = \frac{V_d}{V} = \frac{V_d}{V_0} \times \frac{V_0}{V} = \Phi_0 \times \frac{V_0}{V} \tag{3}$$

where $V_d$ is the volume of the gel in the dried state, $V$ is the volume of the gel when it reaches swelling equilibrium, and $V_0$ is the volume of the gel in the reference state (as-prepared state). For the same gel sample, we can assume that the volume fraction of the gel in the reference state $\Phi_0 = V_d/V_0$ is constant, thus we can obtain $\Phi/\Phi_0 = V_0/V$.

For the gel matrix swelling in the deionized water, the gel swells significantly so that its volume is 12.5 times larger than its original volume ($\Phi/\Phi_0 = 0.08$) when it reaches the swelling equilibrium, leading to the decrease of the elastic modulus G′ after swelling in the deionized water.

For the gel composites with PDA inside, the as-prepared gel composites were swelled by either deionized water or PDA solutions, whose concentrations are the same as that inside the gel composites. The gel composites were swelled until they reached the swelling equilibrium. So, one sample was measured in its three states: as-prepared state, swelling by deionized water, and swelling by PDA solutions, and they are named as As-prepared, $H_2O$ swelling, and PDA swelling, respectively. There are three factors influencing the elastic modulus of the gel composite: the volume fraction of the gel, the number of chemical bonds, and the number and strength of the physically ionic bonds. Here one sample in its three states will have the same chemical bonds, so their viscoelastic properties were determined by the synergetic/antagonistic contribution from the physical bonds and their volume fractions.

Dynamic frequency sweeps at different molar charge ratio $r$ for the gel composites with LPDA inside are shown in Fig. 4b–d. For $r = 0.3$ with LPDA, when the as-prepared sample is swelled by water, the volume increases 4 times such that the volume fraction decreases to $\Phi/\Phi_0 = 0.25$ (see below). For $r = 0.3$, the number of physically ionic bonds arising from the complexation between the LPDA and gel matrix is small. Therefore, although swelling by water can remove some small ions and lower the ionic strength inside the gel composite due to the Donnan equilibrium and the physically ionic bonds can be strengthened, the decrease of the volume fraction dominates the elastic modulus as $G'\sim\Phi^{1/3}$, thus G′ decreases roughly to half compared to that of the as-prepared state (Fig. 4b). However, for the sample with $r = 0.3$ swelled by LPDA solution whose concentration is the same as that inside the gel composite, G′ increases due to the synergetic contribution from the physically ionic bonds and the volume fraction increase. On one hand, the volume shrinks to $\Phi/\Phi_0 = 2.4$ after swelling by LPDA solution (see below), leading to the increase of G′. On the other hand, $r = 0.3$ is below the saturation of complexation with the gel matrix, so that when the gel composite is swelled by LPDA solutions, more LPDA chains in solution will diffuse inside the gel composite and complex with the unoccupied charged sites on the gel matrix. Please note that the electrostatic complexation between the LPDA and the unoccupied charged sites on the gel matrix is the driving force to make the LPDA chains diffuse inside the gel composites. So, the increase of the number of physically ionic bonds can also contribute to the increases of G′. Moreover, G′ and G″ have a stronger frequency dependence (larger exponent $n$, $m$ values) compared to that of the as-prepared state (Fig. 4b), indicating the gel composite transfer from the chemical gels to more like the physical gels with ionic bonds. Therefore, after swelling by LPDA solutions for gel composite with r = 0.3, the synergistic contribution from the increase of both the number of the physically ionic bonds and the increase of the volume fraction makes the elastic modulus G′ increase 4 times of that for the as-prepared state (Fig. 4b).

For $r = 0.6$ swelled by water, the volume fraction almost keeps constant $\Phi/\Phi_0 = 1.04$ (see below). But the physically ionic bonds get strengthened when small ions are removed when swelling by water so that G′ increases more than 10 times and has a stronger frequency dependence (larger exponent $n$ value) compared to G′ of the as-prepared state (Fig. 4c). While for $r = 0.6$ swelled by LPDA solution, G′ is only slightly larger than that of the as-prepared state due to the antagonistic effect of volume shrinkage and the increase of the ionic strength in the system (Fig. 4c). For the apparent charge ratio $r = 0.6$, we can roughly estimate that the actual molar charge ratio of LPDA/gel matrix is 0.79 if the monomer chemical conversion is considered. It is close to the stoichiometric complexation, and further swelling by LPDA solution whose concentration is the same as that inside the gel

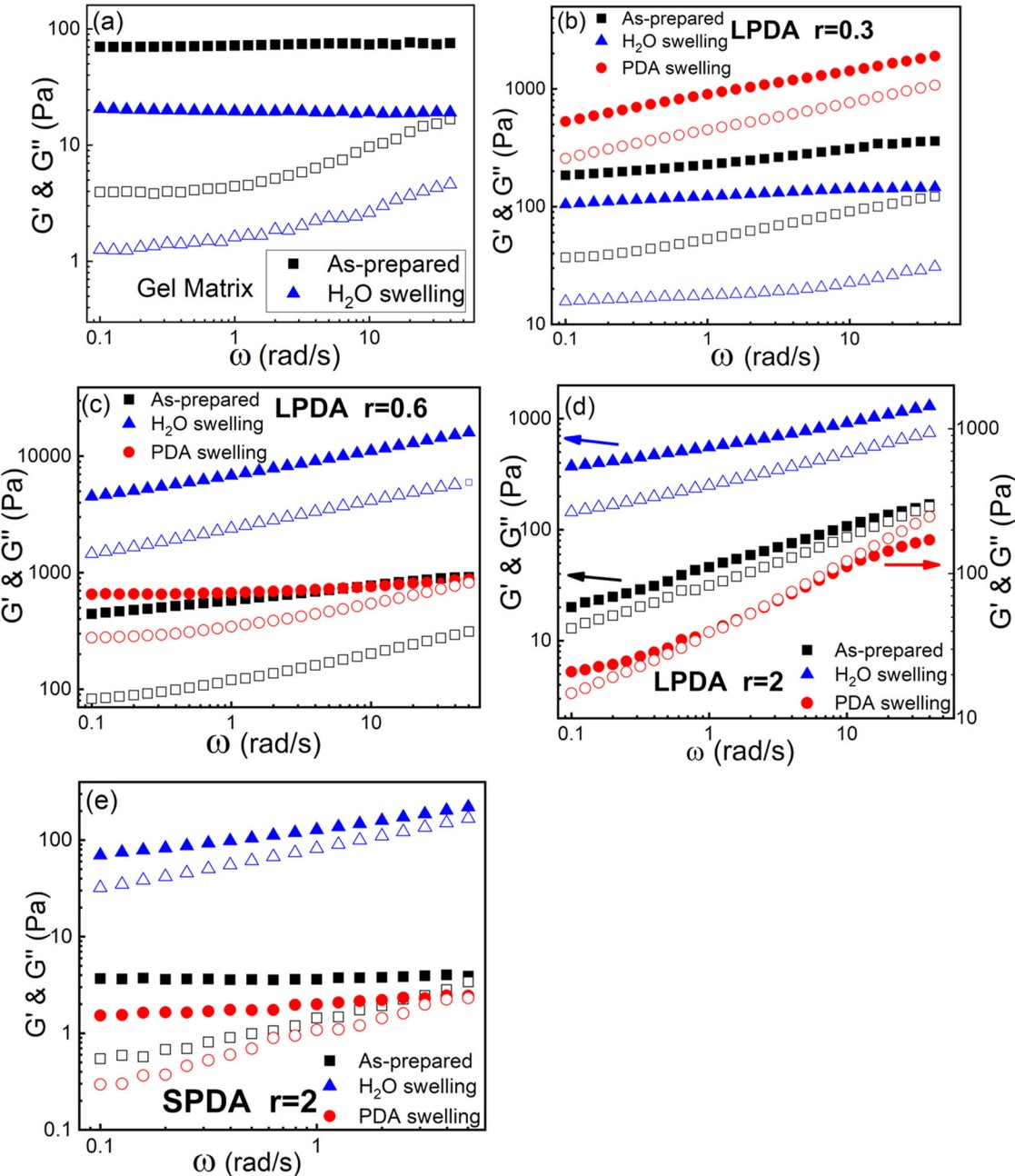

**Fig. 4 | Linear viscoelastic properties of gel composites at different swelling states.** Dynamic frequency sweeps for **a** gel matrix, **b**–**d** Gel composite with LPDA inside with different molar charge ratio $r$, **e** Gel composite with SPDA inside with $r = 2$. Three states were measured for each sample: as-prepared state, swelled by water, and swelled by PDA solutions, whose polymer concentration is the same as that inside the gel composite. Storage moduli G' (closed symbols) and loss moduli G" (open symbols) versus frequency at shear strain $\gamma = 1\%$. The y-axis ranges are important to note. The data in (**d**) for PDA swelling is plotted with separated right y axial for better clarity.

composite may lead to oversaturated complexation and more ions being introduced inside the gel composite.

For $r = 2$, there are much more guest LPDA molecules are present beyond the saturation of complexation with the gel so that the extra uncomplexed LPDA chains and their counterions will increase the ionic strength of the gel composite, thus weaken the physically ionic bonds between the guest and host, leading to an order of magnitude decrease of G' at $r = 2$ compared to that at $r = 0.6$ for the as-prepared state (Fig. 4c, d). When swelling by LPDA solutions, the moduli decrease even further, and a solid-liquid transition even shows up at $\omega = 2$ rad/s for $r = 2$ (Fig. 4d) due to the salt-induced softening saloplasticity effect[55]. Although the samples with $r = 2$ are so soft that their volumes are hard to precisely measure, the volume does not change much after swelling. On the other hand, for $r = 2$ swelled by water, G' increases more than 10 times compared to that of the as-prepared state (Fig. 4d) due to the decrease of the ionic strength in the system after swelling by water, thus the physically ionic bonds are strengthened. Dynamic frequency sweep of the gel composites with shorter guest chains SPDA at $r = 2$ is also shown in Fig. 4e as a representative. Although they do not exhibit a solid-liquid transition, the moduli are much smaller and G' has less frequency dependence compared to that of the gel composites with LPDA at $r = 2$. The impact of the guest chain size (or $M_w$) on the viscoelastic properties of the gel composites will be discussed in the next section.

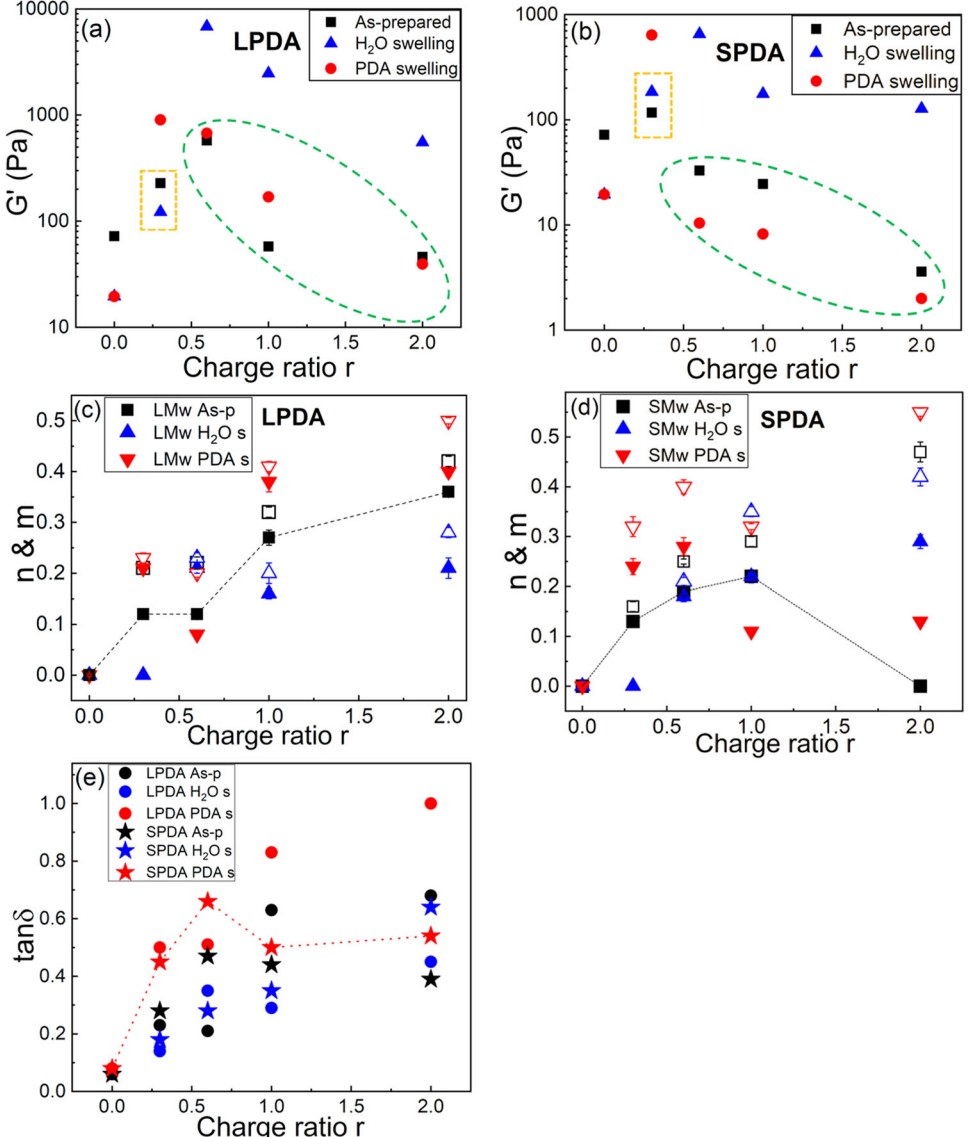

**Fig. 5 | Map of elastic modulus and viscoelastic properties in the PDA-gel matrix complexation.** All the data were obtained from dynamic frequency sweeps at ω = 1 rad/s. **a**, **b** Elastic modulus G', **c**, **d** fitting values of *n* (filled symbols), *m* (open symbols) (G'-ωⁿ, G"-ωᵐ), and **e** tanδ values (tanδ = G"/G') at different molar charge ratio *r* for gel composites with LPDA and SPDA respectively. Three states were measured for each sample: as-prepared state, swelled by deionized water, and swelled by PDA solutions, whose PDA concentration is the same as that inside the gel composite. s is shorten for swelling and As-p is shortened for As-prepared in all the figures. Each of the data points denotes the average of three individual replicates, and error bars are ± sd.

## Building up a viscoelastic map by tailoring the topological correlated structures of the complexation gel composites

Combined all the data for the gel composites with LPDA, MPDA, and SPDA, we can quantitively tailor the elastic modulus G' (all obtained at ω = 1 rad/s) four orders of magnitude in the range of 1–10⁴ Pa, which are applicable to many biomaterials such as vitreous humor, mucus, nucleus pulposus, etc. Therefore, a viscoelastic map has been built up for the complexation gel composites. As mentioned above, the guest chains with three different chain lengths have the averaged radius of gyration $R_g$ = 47.2 nm, 38.3 nm, and 15.8 nm for LPDA, MPDA, and SPDA, respectively. And when the guest chains are inside the host gel matrix with the averaged mesh size 21 nm, if calculated in three-dimensional space one guest chain can cover 47 meshes, 25 meshes, and <2 meshes for LPDA, MPDA, and SPDA respectively in the host gel matrix. If one chain can cover tens of gel meshes (LPDA and MPDA), it will be able to cooperatively complex with the gel strands from multiple gel meshes and form a topologically quasi-interpenetrating complexation network. While if the chain size is smaller than the mesh size (SPDA), the polycation chains will only complex with the gel strands within each individual gel mesh, thus the single gel strands would become complexed double gel strands (Fig. 1a–d). Such entropy-driven topological correlated structures of the gel composites can lead to entirely different viscoelastic properties.

A summary of the elastic modulus G' (obtained at ω = 1 rad/s) as a plot of charge ratio *r* in their three swelling states of As-prepared, H₂O swelling, PDA swelling for LPDA and SPDA is shown in Fig. 5a, b. The data trend of MPDA is similar to that of LPDA (Supplementary Fig 3a–c). For the as-prepared gels, the ionic strength of the system is in the range of 0–0.4 M due to counterion release, depending on the molar charge ratio *r*. The elastic modulus G' of the gel composites with LPDA, which can form the topologically quasi-interpenetrating complexation network, is overall larger than G' of the gel composites with SPDA under the same condition, indicating that the topologically

quasi-interpenetrating complexation structure can strengthen the gel composites.

For $r = 0.3$ swelled by water, the sample volume with LPDA swells four times such that the volume fraction decreases to $\Phi/\Phi_0 = 0.25$ (see below), while the sample volume with SPDA shrinks ($\Phi/\Phi_0 = 1.36$). One LPDA chain can cooperatively cover and interpenetrate multiple meshes so that it has a heterogeneous distribution inside the gel matrix when $r = 0.3$, which is below the saturation complexation point. Therefore, the gel composite will swell dominated by the Donnan equilibrium, thus making G′ decreases compared to the as-prepared state (orange dashed rectangles in Fig. 5a). However, for sample with SPDA at $r = 0.3$, the chain size is smaller than the mesh size so that SPDA chains can be uniformly distributed in the gel matrix, and complex with the gel strands and strengthen the gel strands within each individual mesh, making the single gel strands become complexed double gel strands. So, for sample with SPDA at $r = 0.3$, complexation effect dominates over the Donnan equilibrium, such that the sample volume shrinks and G′ increases compared to the as-prepared state (orange dashed rectangles in Fig. 5b).

As the molar charge ratio r increases from 0.6 to 2, G′ of the gel composites with both LPDA and SPDA swelled by water are all higher than G′ of their as-prepared state due to the decrease of the ionic strengths when swelled by water, leading to stronger complexations and higher G′. However, G′ values of the gel composites with SPDA swelled by SPDA solutions are all lower than G′ of their as-prepared states, but G′ of the gel composites with LPDA swelled by LPDA solutions are slightly higher than G′ of their as-prepared states (green dashed oval in Fig. 5a, b). Such differences arise from different conformational entropy of the guest chains, leading to different topological correlated structures between the guest chains and the host gels. When the gel composites with LPDA swelled by LPDA solutions, although some small ions will come inside the gel composites, so that the ionic strength inside the gel composites increases and the complexation between the guest and host can be partially screened and weakened, the topological correlations from long-chain connectivity and large conformational entropy can make one LPDA chain cooperatively cover and interpenetrate around 47 meshes and form a topologically quasi-interpenetrating network, which dominates the elastic modulus of the gel composites, making G′ slightly higher or similar to that of their as-prepared states at higher r. Besides, the volume fractions of the gel composites with LPDA swelled by LPDA solutions almost keep constant at different r due to the existence of topologically quasi-interpenetrating network (see below). Therefore, such a topologically quasi-interpenetrating network due to large conformational entropy can stabilize the elastic modulus and volume fractions of the gel composites.

On the other hand, for gel composites with SPDA swelled by SPDA solutions, when the gel matrix is fully complexed with SPDA chains at higher r, which is above the saturation of complexation point, the extra uncomplexed free SPDA chains can fill the voids of each fully complexed uncharged gel meshes. These extra free SPDA chains can act as a plasticizer, they repel each other within the voids of each individual gel meshes due to the excluded volume interactions and electrostatic repulsion between similarly charged SPDA chains, making the gel meshes swelling. This causes a decrease in the network stiffness and volume fractions. We have defined such a state as repulsive gel composite. Besides, the extra SPDA chains and their counterions will increase the ionic strength of the system, thus weaken the physically ionic bonds and can also contribute to the decrease of G′. Therefore, for gel composites with SPDA swelled by SPDA solutions, G′ values decrease with r and are all lower than that of the as-prepared state at higher r (green dashed oval in Fig. 5b), and their volume fractions decrease continuously as r increases from 0.3 to 1.0 (see below).

Moreover, for gel composites with LPDA, as $r$ increases, both G′ and G″ have a stronger frequency dependence (G′-$\omega^n$, G″-$\omega^m$), and the corresponding n and m exponents increase (Fig. 5c), indicating the gel composites gradually transfer from the chemical gels to more like physical gels as $r$ increases. For both LPDA and SPDA gel composites swelled by water, $n$ and $m$ exponents increase with $r$ due to stronger complexations, so that the system will transfer from chemical gels to physical gels with ionic bonds with increasing $r$. For samples in the As-prepared and PDA swelling states, LPDA and SPDA gel composites behave differently. For LPDA gel composites, $n$ and $m$ values increase with $r$, while for the SPDA gel composites, $n$ values first increase and then decrease. Because when the PDA concentration is beyond the saturation of complexation at higher r, the ionic strength increases with $r$ and the complexation is weakened, but the LPDA gel composites still behave like the physical gel due to the topologically quasi-interpenetrating network arising from the large conformational entropy. While for the gel composites with SPDA, when the SPDA concentration is beyond the saturation of complexation at higher $r$, they become repulsive gel composites, and the complexation insides the gel composites are also weakened due to the increase of the ionic strength with $r$, so that at higher $r$ G′ have less frequency dependence and smaller $n$ values with $r$, and $n$ even decrease to 0 at $r = 2$ for the as-prepared state (Fig. 5d), indicating that repulsive interaction dominates the repulsive gel composite, which behaves similar to the repulsive glass[63,64]. However, (m-n) values of the repulsive gel composites for the SPDA gel composites at $r = 2$ in three states are much larger compared to the (m-n) values of LPDA gel composites at $r = 2$. Larger (m-n) values indicate the sample is more elastic in nature, and smaller (m-n) values indicate it is physical gels with ionic bonds. Therefore, for the SPDA gel composites at higher $r$ they become a repulsive gel composite with more elasticity, compared to the LPDA gel composites which have topologically quasi-interpenetrating network.

Besides, tanδ values (obtained at ω = 1 rad/s) show similar trend to $n$, $m$ values for both LPDA and SPDA gel composites (Fig. 5e). For both the LPDA and SPDA gel composites swelled by water, tanδ increases with r due to stronger complexation, so that they can dissipate more energy with stronger ionic bonds. For samples in the As-prepared and PDA swelling states, tanδ of LPDA gel composites increases with $r$, while tanδ of the SPDA gel composites first increases and then decreases due to the repulsive gel composites being more elastic in nature. While the topologically quasi-interpenetrating network makes the LPDA gel composites have better-damping properties, thus tanδ increases with $r$. Please note that the highest tanδ values occur at LPDA gel composites in the PDA swelling state at higher $r$, demonstrating that quasi-interpenetrating network made of long LPDA chains complex with the gel matrix can make the system dissipate more energy and have better-damping properties, and their viscoelastic properties will incline to more viscous rather than elastic.

Overall, because of various topologically correlated structures of the complexation gel composites, for gel composites with LPDA, as $r$ increases, they will have the chemical gels to physical gels with ionic associations to topologically quasi-interpenetrating network structural transition. While for gel composites with SPDA, as $r$ increases, they will have the chemical gels to physical gels with ionic bonds to repulsive gel composites structural transition. Based on various topologically correlated structures between the guest and host, the viscoelastic properties of the complexation gel composites can be quantitatively tuned in a broad range, ranging from tough gel to ultrasoft gel, and from elastic-like gel with low damping properties to viscous-like gel with high damping properties, so that a viscoelastic moduli map, which is applicable to many soft biomaterials such as vitreous humor, mucus, nucleus pulposus, etc., can be built up for the complexation gel composites. The gel charge density investigations to tune the moduli dependency on the ionic strength and studies for a similar gel system under physiologic conditions are relegated to future work. The applications of our physical model and mechanism need

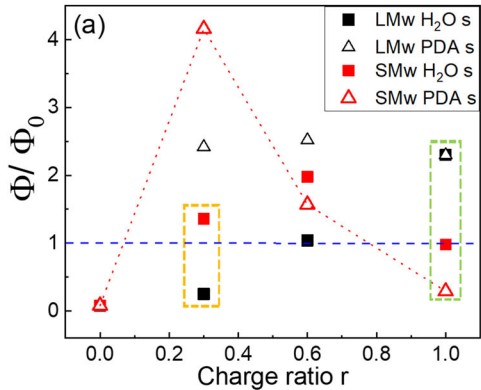
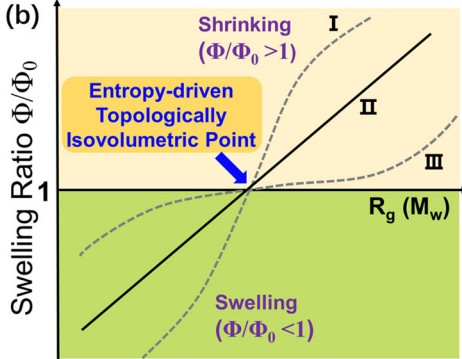

**Fig. 6 | Swelling measurements and the proposed Entropy-driven Topologically Isovolumetric Point for the complexation gel composites. a** The swelling ratio $\Phi/\Phi_0$ at different r for gel composites with LPDA and SPDA swelled by water and PDA solutions, whose polymer concentration is the same as that inside the gel composite. $\Phi_0$ is the volume fraction of the as-prepared sample and $\Phi$ is the volume fraction of the samples at swelling equilibrium. s is shorten for swelling.

**b** Schematic illustration of swelling ratio $\Phi/\Phi_0$ as a function of $M_w$ or chain size $R_g$ of the guest PDA chains inside the gel composites swelled by water at fixed r beyond the complexation saturation point. We predict an Entropy-driven Topological Isovolumetric Point at which the gel composites neither swell or shrink ($\Phi/\Phi_0 = 1$) at swelling equilibrium. I, II, and III denote three possible dependent pathways for the swelling ratio $\Phi/\Phi_0$ as a function of guest chain size $R_g$.

specific requirement for the materials. For example, an approach to repairing cartilage is to permanently patch an injury or replace damaged tissue with materials which are a non-degradable and non-toxic, and polyacrylamide and PDA used here are non-degradable and non-cytotoxic in vivo[65–68].

### Discovery of the Entropy-driven Topologically Isovolumetric Point and the swelling ratio investigations

Figure 6a shows the volume change for the LPDA and SPDA gel composites swelled by PDA solutions and water, respectively. The volume change of MPDA gel composites is similar to that of LPDA (Supplementary Fig 3d). For the charged gel matrix swelled by water ($r = 0$), the gel swells 12.5 times compared to its original volume ($\Phi/\Phi_0 = 0.08$) due to the osmotic pressure and Donnan equilibrium. For the PDA-gel composites swelled by water, on one hand, both LPDA and SPDA gel composites tend to swell due to the osmotic pressure and Donnan equilibrium. On the other hand, the complexation due to the ionic bonds can make the gel composites tend to shrink. For LPDA gel composite, the long LPDA chains can cooperatively cover and complex with tens of gel meshes so that the LPDA chains can hold multiple gel meshes together to make the network shrink. So for the LPDA gel composites swelled by water, the gel volume shrinks with $r$ due to more complexation sites being introduced to form the topologically quasi-interpenetrating network. While for the LPDA gel composites swelled by LPDA solutions, although the complexation is weakened by introducing more small ions inside the gel composites when swelled by LPDA solutions, the topologically quasi-interpenetrating network structures will make their volume fractions almost keep constant with $r$. For the SPDA gel composites swelled by SPDA solutions, complexation dominates at $r = 0.3$ so that the volume fraction increases. However, when $r$ is further increased, the volume fraction decreases with r due to the repulsive gel composite structure, which arises from the excluded volume interactions and electrostatic repulsion of the uncomplexed SPDA chains inside each individual gel meshes. Therefore, at $r = 1$ swelled by PDA solutions, LPDA gel composite shrinks while SPDA gel composite swells (green rectangle in Fig. 6a). For the SPDA gel composites swelled by water, the volume fraction first increases with $r$ due to more complexations being introduced, and then decreases to $\Phi/\Phi_0 = 1$ at $r = 1$ dominated by the repulsive gel composite structure.

Based on the preliminary results in Fig. 6a, we found two isovolumetric points where the volume neither swells nor shrinks ($\Phi/\Phi_0 = 1$) at swelling equilibrium. At $r = 1$ swelled by water, $\Phi/\Phi_0$

increases as $M_w$ increases (green rectangle in Fig. 6a). Therefore, we have designed a new strategy to tune the swelling ratio of the complexation composite by only changing the conformational entropy and their topologically correlated structures. Here we have proposed that for the complexation gel composites swelled by water at fixed r, which is beyond the complexation saturation point, by only tuning the chain size $R_g$ (or $M_w$) of the guest PDA chains in the fixed host gel matrix, the gel composites are able to realize the repulsive gel composite to semi-interpenetrating network to topologically quasi-interpenetrating network structural transition and their volume will change from swollen state to shrunk state as $R_g$ increases. So, we can precisely figure out an isovolumetric point ($\Phi/\Phi_0 = 1$) by only tuning the guest chain size and the conformational entropy, as schematically illustrated in Fig. 6b. Such an isovolumetric point is solely controlled by the large conformational entropy and topologically correlated structures between the guest chains and host gels in the complexation systems. Therefore, we have defined it as an Entropy-driven Topologically Isovolumetric Point for the complexation gel composites at the swelling equilibrium. The three curves in Fig. 6b show three possible dependencies for the swelling ratio $\Phi/\Phi_0$ as a function of guest chain size $R_g$. In this model system, swelling ratio can be tuned from highly shrinking to isovolumetic to highly swelling. More importantly, the viscoelasticity and swelling can be independently controlled in our complexation gel composites, so that they can fit in various biomedical applications. For example, high-swelling hydrogels are for tissue engineering and drug delivery due to their high cell recruitment and migration and ability for molecules diffusion and release. Non-swelling hydrogels are favored for tissue engineering and bioelectronic due to their mechanical properties and stability, but lacking ability to load and release drug. Shrinkable hydrogels show great potential for scaffold as their stiffness and promotion for cell attachment[69]. Tuning the swelling ratio solely by utilizing conformational entropy and topologically correlated structures may provide a new way to control the swelling ratio of charged gel composites. A systematic study of the proposed Entropy-driven Topologically Isovolumetric Point is relegated to our future work.

### Discussion

In the complexation hydrogel composites, by utilizing topologically frustrated dynamical state and changing the size ratio $R_g/\xi$ of the guest over the host, the molar charge ratio $r$, and the swelling methods, their topological correlated structures between the guest and host can be precisely tuned. For gel composites with LPDA, as $r$ increases they will

have the chemical gels to physical gels with ionic associations to topologically quasi-interpenetrating network structural transition. And for gel composites with SPDA, whose chain size is smaller than the mesh size, it becomes a repulsive gel composite when $r$ is above the saturation complexation point due to the excluded volume interactions and electrostatic repulsion of SPDA chains. Based on various topologically correlated structures, the viscoelastic properties of the complexation gel composites can be quantitatively tuned in a broad range, ranging from tough gel to ultrasoft gel, and from elastic-like gel with low damping properties to viscous-like gel with high damping properties. Therefore, a viscoelastic moduli map with G′, $n$, $m$ exponents and tanδ values is built up. Such a moduli map with G′ in the range of $1\text{-}10^5$ Pa is applicable to many soft biomaterials and tissue engineering. Moreover, we have proposed the concept of Entropy-driven Topologically Isovolumetric Point, where the volume does not change at swelling equilibrium. Tuning the swelling ratio solely by utilizing conformational entropy and topologically correlated structures may provide a new strategy to control the swelling ratio of the charged complexation gel composites. Our findings provide fundamental understanding of the relationship between entropy-driven correlated structures and their viscoelastic properties of the complexation hydrogel composites and may inspire further research on governing other physical properties by utilizing the entropy effect in the charged hydrogel systems.

## Methods
### Instrument
The $^{13}$C NMR spectrum was recorded on a Bruker NEO 700 spectrometer with cryo BBO probe. The shear modulus experiment was performed with a stress-controlled rheometer (Anton parr MCR 502), using a 25 mm roughened parallel plate geometry. The frequency sweeps were performed with a shear strain of γ = 1%, which is in the linear viscoelastic region. The light scattering was performed on a laser light scattering spectrometer (ALV/CGS-3) equipped with a multi-τ digital time correlator (ALV-7004) and a laser light source with wavelength of 532 nm. Dynamic light scattering measures the intensity-intensity time correlation function $g_2(q, t)$ by means of a multi-channel digital correlator and related to the normalized electric field correlation function $g_1(q, t)$ through the Siegert relation[70]. For each sample, the intensity at each of the scattering angles 30°, 40°, 50°, 60°, 70°, and 90° were correlated, and the relaxation time averaged for three different spatial locations within the gel samples. Both CONTIN method and multiple exponential fitting method were used to analyze the data[70]. All the deionized water was obtained from a Milli-Q water purification system (Merk Millipore IQ 7000). The resistivity of deionized water used was 18.2 MΩ·cm.

### Sample preparation
The complexation gel composites, which are composed of poly (acrylamide-co-sodium acrylate) gel with 80% charge density and 1.5% crosslinking density, and with the poly (diallyldimethylammonium chloride) (PDA) chains embedded inside, were synthesized by free radical polymerization. 0.383 g sodium arylate, 0.076 g arylamine, 12 mg Bis (acrylamide) were mixed with certain amount of concentrated PDA solutions, then deionized water was added until the total volume is 10 mL. Then the pre-gel solution was bubbled with nitrogen gas for 20 min to remove any dissolved oxygen. To initiate the polymerization, 15 μL of TEMED and 5 mg of ammonium persulfate dissolved in 0.2 mL of water were added to initiate the polymerization. The chemical reaction continued for 48 h at room temperature. For the light scattering samples, the pre-gel solution was filtered by the 220 nm PVDF hydrophilic filter into a light scattering tube to remove the dust. The light scattering tubes were cleaned by using distilled acetone through an acetone fountain setup before use.

## Data availability
All data that support the findings of this study are available from the corresponding author upon request.

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

## Acknowledgements

This work was supported by the National Key R&D Program of China (Grant No. 2023YFE0124500), the National Natural Science Foundation of China (Grant No. 22273114), the National Key R&D Program of China (Grant No. 2023YFC2411203), and International Partnership Program of the Chinese Academy of Sciences (Grant No. 027GJHZ2022061FN). We thank Dr Jun Feng Xiang for assistance with NMR measurements.

## Author contributions

D.J. designed the system, and G.K.W. synthesized the samples and performed all the experiments. D.J., G.K.W., and Y.M.Y. discussed and contributed to the interpretation of the data and writing the manuscript. The project was supervised by D.J.

## Competing interests

The authors declare no competing interests.
