## [Peer Review File · Nature Communications]

Programming viscoelastic properties in a complexation gel composite by utilizing entropy-driven topologically frustrated dynamical stateReviewers' Comments:

Reviewer #1:

Remarks to the Author:

"Programming viscoelastic properties in a complexation gel composite by utilizing entropy-driven topologically frustrated dynamical state" is an interesting manuscript that touches upon a popular aspect of charged polymer complexation, but now following on the author's prior efforts in topological trapping. Prior efforts discovered and theoretically explain a new transport regime when the polymer is much larger than the gel apparent mesh size that leads to topological pinning and non-diffusive behavior. The present study branches out to use complexation in addition to the topological confinement. They find some interesting rheological consequences in the hydrogels that provides a unique contribution to the field that are more focused on solutions of oppositely charged mixtures of linear chains or block copolymers. The manuscript describes the different samples of negatively charged anion gel polymerized in the presence three guest polycations. The data quality look very good and the overall content is novel and can be improved or clarified through the review process. I do appreciate that this is a complicated topic where the experiments requires carefully preparation and attention to detail as these are not the type of measurements that lend to trivial automation.

Methodology, experimental and data analysis detail

- Reporting PDA "Mw = 400-500KDa" is not as informative as providing either absolute Mw from light scattering, relative values from GPC, etc. Further, kilo is lowercase "k" use kDa or kg/mol.
- What is the polydispersity (Mw/Mn or relative values from SEC, or estimated values form DLS) of the PDAs as synthesized and does this affect how much polyelectrolyte can be confined? Cite the relevant papers for the procedures if used directly from the literature.
- General comment. Are the error bars smaller than the size of the symbols? For example what are the error bars in the curve fit values of m, n ? Error bars should be reported to understand if trends are meaningful.
- I can follow the author's approach to describe properties across the different PDA content as a function of "r." However, it becomes clear that "r" is only an effective value. The author should explain the reliability in this calculated "r" as there may be rejection of the free polymer from the network, dialyzed out, or side reactions. A brief comment on how the concentrations are measured would be appreciated and comparison to how other hydrogel platforms such as double-network gel, or the interpenetrating hydrogel determine concentration.

Significance to the field and related fields

- The vitreous humor, mucus, nucleus pulposus were examples of soft biomaterials given in the introduction. The authors could increase the manuscript readership by showing, in a visual manner, how their "programmable" properties compare to such soft biomaterials. One typically finds an Ashby plot of properties such as modulus versus density, for example. A recent example can be found here <https://pubs.acs.org/doi/10.1021/acsami.3c04331> What properties are the author's comparing exactly to the aforementioned soft biomaterials? How do the properties compare to other synthetic hydrogels formed by different polymer architectures such as tetraPEG gels, bottle brush gels, double networks, IPN, etc...By explaining this more, the manuscript may highlight the gaps in polymer materials properties achievable by this approach.
- The tunability of the tan delta seems very interesting, but the measurement seems to lack some context. The data transition from more elastic to viscous as r increases. Please see my prior question as there needs to be a connection to the broader materials property attributes. The observation are likely related to rheology in polyelectrolyte complexes (reviews are available, by Larson for example) where a much wider polymer concentration can be studied and the effect of salt concentration. Few have examined the effect of concentration and molecular weight asymmetry.

Introduction and Results and Discussion clarifications and Corrections

- The authors are making an equivalence between the DLS interpreted hydrodynamic mesh size and the static mesh size, this is not proven for polymer gels (201-207). The generalized Stokes Einstein relationship was found to apply to polyelectrolyte complex solutions, recently, such that the zero-shear solution viscosity, not the solvent viscosity, provides a more accurate interpretation of the cooperative diffusion coefficient for the hydrodynamic correlation length. In the present case, the solution viscosity may not be far from the solvent within a small factor depending on the concentration. Only one data point was brought up, but this was not shown for the different degrees of crosslinking or concentration in the present system.
- The term "Entropy driven topological isovolumetric point" was coined, however, I don't really see the data where no gel swelling occurs. I can see the boxes that show a trend between LPDA and SPDA and some points are at a swelling ratio of 1. A no swell or no shrink gel is very interesting, it would be more convincing of a full trend with more than two molecular weights were shown. Figure 6a does not show the MPDA (middle molecular weight data).
- Line 372 the authors refer to a close to stoichiometric complexation at an r value $r=0.60$. Please explain or clarify as by the author's own definition of "r" is the ratio of the charges determined by the monomer charge and concentrations.
- What is the meaning of the dotted circles in Figure 5a and 5b?
- What are the authors trying to explain with Figure 5c and 5d?
- Authors attribute an "enthalpy effect" for complexation on line 355 to 358. Provide a citation as to who determined this to be true as it is not universally accepted.

Reviewer #2:

Remarks to the Author:

This manuscript presents a new class of hydrogel materials wherein viscoelasticity and swelling can be independently controlled. The materials have potential biomaterial applications as their range of shear moduli are similar to several tissues of importance in regenerative medicine. The experimental design, methodology, analysis, interpretation and presentation are outstanding. Concerns primarily relate to weak support for the materials in biomedical applications and some sections of the text being challenging to follow.

Biomedical applications: The polymers selected have potential in biomaterials applications such as cartilage repair in articulating joints and the spine. As mentioned in the manuscript, the hydrogels' shear moduli range is relevant to that of cartilage and vitreous fluid. Cartilage repair has been a particularly important area in regenerative medicine, with top leaders in the biomaterials field (Anseth, Burdick, Khademhosseini to name a few) investing major research focus over decades on hydrogels for cartilage repair. The reported hydrogels have potential to significantly impact this field. However, several concerns are the following:

1. An approach to repairing cartilage is to permanently patch an injury or replace damaged tissue with a non-degradable material. It is presumed that the reported hydrogels are not degradable under physiologic conditions because polyacrylamide is well known to be non-degradable and a brief literature search suggests that poly(diallyldimethylammonium chloride) (PDA) may be non-degradable as well. It would be helpful for the authors to briefly address this topic.
2. Along similar lines, polyacrylamide (when washed to remove any residual acrylamide monomers) is not cytotoxic and a brief literature search indicates that PDA is potentially antimicrobial but not toxic to mammalian cells. Please comment on this topic as well.
3. Hydrogels exist that span the same range of shear modulus. For example, this reviewer has unpublished data that polyacrylamide gels of varying bis:acrylamide span a shear modulus range of

10-10,000 Pa. While swelling ratios were not measured, massive changes in swelling were visually observed, with swelling inversely proportional to shear modulus. A brief discussion supporting the need for a class of hydrogels with these ranges of tunable shear moduli as well as swelling (from shrinking, to isovolumetric, to highly swelling) in biomedical applications would significantly strengthen the rationale for this manuscript.

4. For biomedical applications, physiologic conditions are the most relevant for characterization. Typically, materials are evaluated in phosphate buffered saline (PBS). Is there any way to relate the findings of this work to a physiologic ionic strength? This is of particular concern for the gels equilibrated in water, as the shear moduli of these gels range 100-10,000 Pa, yet samples "as prepared" (not equilibrated in a solution following polymerization) and samples equilibrated in PDA solutions only range 10-1000 Pa.

Other concerns: The manuscript is highly enjoyable to read because of its clear writing and appropriate level of relatively complex theory (motivation, explanation and use in analysis). However, there are some areas that should be addressed in order to yet improve the ease of reading of the work.

5. The paragraphs are very long (and will appear yet longer when formatted for Nature Communications), making them hard to follow. It is highly recommended to break up paragraphs -- in areas such as,

- Line 119, where the introduction transitions from background to description of the work herein
- Line 493, where results of m and n transition to $\tan(\delta)$

These are but two examples that were of note to this reviewer, but it is recommended that consideration should be given to breaking up most paragraphs throughout.

6 . Extremely minor issue: line 42 states "rad/s⁵" (where 5 is a citation) but at first it was read to be a typo suggesting rad/s². Not sure what can be done here, but wanted to mention it because rad/s² would be confusing.

7. Line 45: "modulate their moduli" is repetitive.

8. In lines 158-159, LPDA, MPDA and SPDA are defined. On first reading, it was assumed that L stands for low, M stands for medium and S was unclear. When noticing the MW ranges for each acronym, it was then assumed that L stands for "large" and M stands for "medium" and S stands for "small." This notation is fine, but because ranges are typically ordered as low→high, it is suggested that the acronyms are listed as such and defined.

9. Lines 259-262 contain essential information towards interpretation of the results. By breaking up the paragraph or another means, it would be helpful if these sentences were more obvious.

10. Line 362: Is "synergistic" meant instead of "synergetic"? They might mean the same thing, but synergistic is more commonly used.

12. For Figure 4a-d, it would be helpful to point out in the caption or text that the y-axis ranges are important to note.

13. Line 399 is confusing. Are the values reported in Figure 4d actual/2.5? Please clarify this normalization and substitute "shifted" for a more specific term.
14. All figures: The placement of (a), (b), etc. varies throughout. It would be helpful if they were consistently located in the same corner of each panel throughout.
15. Line 487: It is suggested that "besides" be replaced with "in contrast," "however" or "yet"
16. Figures 5 and 6: Please write out or define "s" and "As-p" in the symbol legend or caption.
17. Figure 6b: What do I, II and III denote? Also, orange letters on the green background are hard to distinguish.

Response to Referee #1

We thank the reviewer for the positive endorsement and appreciation for the careful experiments and our unique contribution to the field that are more focused on solutions of oppositely charged mixtures of linear chains or block copolymers. We are also grateful to the reviewer for several excellent comments. Those comments and suggestions have improved the quality of the manuscript significantly. We have addressed all points raised by the reviewer as detailed below and also in the revised manuscript. All the changes are highlighted in the revised manuscript. The comments of the reviewer are in black and our response is in blue.

1. • “Programming viscoelastic properties in a complexation gel composite by utilizing entropy-driven topologically frustrated dynamical state” is an interesting manuscript that touches upon a popular aspect of charged polymer complexation, but now following on the author’s prior efforts in topological trapping. Prior efforts discovered and theoretically explain a new transport regime when the polymer is much larger than the gel apparent mesh size that leads to topological pinning and non-diffusive behavior. The present study branches out to use complexation in addition to the topological confinement. They find some interesting rheological consequences in the hydrogels that provides a unique contribution to the field that are more focused on solutions of oppositely charged mixtures of linear chains or block copolymers. The manuscript describes the different samples of negatively charged anion gel polymerized in the presence three guest polycations. The data quality look very good and the overall content is novel and can be improved or clarified through the review process. I do appreciate that this is a complicated topic where the experiments requires carefully preparation and attention to detail as these are not the type of measurements that lend to trivial automation.

Response: We are grateful for the warm endorsement on novelty and unique contribution of our findings to the field that are more focused on solutions of oppositely charged mixtures of linear chains or block copolymers. We also thank the reviewer for the succinct summary of our previous and current work and for appreciating our careful measurements and attention to details. Now the quality of the manuscript has improved significantly through the review process.

Methodology, experimental and data analysis detail

2. • Reporting PDA “Mw = 400-500KDa” is not as informative as providing either absolute Mw from light scattering, relative values from GPC, etc. Further, kilo is lowercase “k” use kDa

or kg/mol.

Response: We thank the reviewer for the good question. The information of “M_w=400-500kDa” is provided by the Sigma-Aldrich company, and we agree with the reviewer that this is not informative. Actually, in this work the key physical parameter to tune the topological correlated structures between the guest PDA chains and host gel network is the size ratio of the PDA polymer chain size R_g over the mesh size ξ of the gel network. Therefore, the averaged PDA polymer chain size, which is denoted by the averaged radius of gyration R_g, was carefully measured by static light scattering (SLS). The averaged chain size R_g are 47.2nm, 38.3nm, 15.8nm for LPDA (M_w=400-500kDa), MPDA (M_w=200-350kDa), and SPDA (M_w <100kDa), respectively (line 237 in the revised version). Besides, PDA chain is charged polymers, so it is very difficult to characterize charged polymers by GPC or by Zimm plot from light scattering. The M_w information provided by the Sigma-Aldrich company was characterized from the neutral PDA polymers (before making them charged), then they make the neutral polymers become charged.

Thanks for the correction. Now we have changed “KDa” into “kDa” throughout the whole manuscript and Supporting Information.

3. • What is the polydispersity (M_w/M_n or relative values from SEC, or estimated values from DLS) of the PDAs as synthesized and does this affect how much polyelectrolyte can be confined? Cite the relevant papers for the procedures if used directly from the literature.

Response: Thanks for the nice suggestion! (1) As the reviewer noticed, the PDA polymers purchased from Sigma-Aldrich company are relatively polydisperse so that only a molecular weight range is provided for LPDA (M_w=400-500kDa), MPDA (M_w=200-350kDa), and SPDA (M_w <100kDa). As suggested by the reviewer, we have now characterized the polydispersity of PDA by using cumulant analysis from DLS data. For a polydisperse sample system, the electric field correlation function g₁(t) can be expressed as a sum or integral over a distribution of decay rates G(Γ) by

$$g_1(t) = \int_0^{\infty} G(\Gamma) \exp(-\Gamma t) d\Gamma$$

where G(Γ) is normalized so that $\int_0^{\infty} G(\Gamma) d\Gamma = 1$. In the cumulant analysis method, we can get

$$g_1(t) = \exp(-\langle \Gamma \rangle t) \left(1 + \frac{\mu_2}{2!} t^2 - \frac{\mu_3}{3!} t^3 + \dots \right)$$

Where $\mu_i = \int_0^\infty G(\Gamma)(\Gamma - \langle \Gamma \rangle)^i d\Gamma$ is the i th cumulant. The relative line width can be obtained from cumulant analysis $\mu_2 / \langle \Gamma \rangle^2 = \int_0^\infty G(\Gamma)(\Gamma - \langle \Gamma \rangle)^2 d\Gamma / \langle \Gamma \rangle^2$, and the polydispersity M_w/M_n can be calculated from the approximate expression $M_w / M_n = 1 + 4\mu_2 / \langle \Gamma \rangle^2$ (Han, C. C.; Akcasu, A. Z. *Scattering and Dynamics of Polymers*, John Wiley & Sons, 2011; Koppel, D. E. *J. Chem. Phys.* 1972, 57, 4814–4820; Wei, G. M *et al. Macromolecules* 2013, 46, 1212–1220)

Here estimated by the cumulant analysis of the DLS data, the relative line width of the LPDA sample ($\mu_2 / \langle \Gamma \rangle^2 = \int_0^\infty G(\Gamma)(\Gamma - \langle \Gamma \rangle)^2 d\Gamma / \langle \Gamma \rangle^2$) is ~ 0.35 , Therefore, its polydispersity can be estimated $M_w / M_n = 1 + 4\mu_2 / \langle \Gamma \rangle^2 = 2.4$, which indicates it is relatively polydisperse, and is consistent with the rang of the molecular weight $M_w=400-500$ kDa provided by the Sigma-Aldrich company. For MPDA, the polydispersity is $M_w / M_n= 2.2$, and for SPDA the polydispersity is $M_w / M_n= 2.3$.

(2) Besides, the relatively polydispersity has little affect on how much polyelectrolyte can be confined. Based on the averaged polymer chain size R_g and the mesh size of the gel network, if calculated in three-dimensional space one guest chain can cover 47 meshes, 25 meshes, and <2 meshes for LPDA, MPDA, and SPDA respectively in the gel network. Therefore, even if there is a polydispersity of the polymer chain size, for LPDA and MPDA they are still able to cover and cooperatively complex with multiple gel meshes, while for SPDA, most of the chains will stay and complex within one individual gel mesh since the chain size of SPDA is smaller than the mesh size of the gel network. The rheological results of LPDA and MPDA are similar, but is entirely different from that of SPDA, which indicates the polydispersity has little influence on the confinement condition of polyelectrolytes and their related rheological results.

Now we have added “The polydispersity M_w / M_n values estimated from cumulant analysis method from DLS data are 2.4 for LPDA, 2.2 for MPDA, and 2.3 for SPDA, respectively⁵⁷⁻⁵⁹.” in line 241-243 in the revised version. And we have also cited three related references 57-59: Han, C. C.; Akcasu, A. Z. *Scattering and Dynamics of Polymers*, John Wiley & Sons, 2011; Koppel, D. E. *J. Chem. Phys.* 1972, 57, 4814–4820; Wei, G. M *et al. Macromolecules* 2013, 46, 1212–1220.

4. • General comment. Are the error bars smaller than the size of the symbols? For example what are the error bars in the curve fit values of m,n ? Error bars should be reported to understand if trends are meaningful.

Response: Thanks for the good suggestion. Error bars have been added in the curve fit values of m,n . Some of the error bars are smaller than the size of the symbols so that they are not visible. All the error bars do not change the trends of the data.

5. • I can follow the author's approach to describe properties across the different PDA content as a function of "r." However, it becomes clear that "r" is only an effective value. The author should explain the reliability in this calculated "r" as there may be rejection of the free polymer from the network, dialyzed out, or side reactions. A brief comment on how the concentrations are measured would be appreciated and comparison to how other hydrogel platforms such as double-network gel, or the interpenetrating hydrogel determine concentration.

Response: Thanks for the comment. We fully agree with the reviewer that the molar charge ratio "r" is an **apparent molar charge ratio**, which is calculated as the ratio of the positively charged monomers of PDA over the negatively charged monomers of the gel matrix in the pregel solutions in feed before the gelation chemical reaction starts. However, as the reviewer suggested, after gelation chemical reaction and dialysis, the true molar charge ratio values depend on several factors, such as the chemical conversion (Fig. 3b), dialyzed out, rejection of the free polymer from the network, degree of ionization of PDA, etc. These factors are very difficult to control so that the true molar charge ratio is very hard to quantify. For other gel systems, for example Gong et al. studied the polyampholyte gel network, which is composed of cationic monomers and anionic monomers, the anion charge fraction f is also defined as the mole of anionic monomers over the total monomers in the pregel solutions in feed before the gelation chemical reaction starts. Meanwhile, they also determined the true anion charge fraction f_{true} by using element analysis. The results show that f_{true} is slightly different from f (T. L. Sun, et al., *Nat. Mater.* 12, 932-937 (2013); X. Li, et al., *Macromolecules* 56, 2, 535–544 (2023)). However, our gel composite system is much more complicated so that it is hard to use element analysis method to determine the true molar charge ratio. For example, the rejection of the free polymer from the network cannot be detected by element analysis method. So we have only used apparent molar charge ratio r , which is the ratio in feed in the pregel solutions.

Now we have clarified this point and changed “the molar charge ratio r ” into “the apparent molar charge ratio r in feed in the pregel solutions” in line 173-174 in the revised version, and added the comment “Please note that although element analysis method was used to determine the true anion charge fraction in polyampholyte gels^{36,45}, our system is more complicated so that it is hard to determine the true molar charge ratio by using element analysis.” in line 175-178 in the revised version, and cited the related references 36, 45: T. L. Sun, et al., *Nat. Mater.* 12, 932-937 (2013); X. Li, et al., *Macromolecules* 56, 2, 535–544 (2023).

Significance to the field and related fields

6. • The vitreous humor, mucus, nucleus pulposus were examples of soft biomaterials given in the introduction. The authors could increase the manuscript readership by showing, in a visual manner, how their “programmable” properties compare to such soft biomaterials. One typically finds an Ashby plot of properties such as modulus versus density, for example. A recent example can be found here <https://pubs.acs.org/doi/10.1021/acsami.3c04331> What properties are the author’s comparing exactly to the aforementioned soft biomaterials? How do the properties compare to other synthetic hydrogels formed by different polymer architectures such as tetraPEG gels, bottle brush gels, double networks, IPN, etc...By explaining this more, the manuscript may highlight the gaps in polymer materials properties achievable by this approach.

Response: Thanks for the excellent suggestion! The values of elastic modulus are compared to that of soft biomaterials. As the reviewer suggested, now we have added a new plot in Fig. 1(g) to make a comparison of our complexation gel composites, synthetic hydrogels, and biomaterials. It clearly shows that our complexation gel composites mainly fill the gap in the ultrasoft-soft regime.

Now we have added this plot in Fig. 1(g) and have added the comment “A comparison of the complexation gel composites in this work with synthetic hydrogels and soft tissues has been made. It clearly shows that our complexation gel composites mainly fill the gap in the ultrasoft-soft regime.” in line 53-56 in the Introduction part in the revised version. We have also cited the references 1-5, 28, 39-44 in Fig. 1(g) caption.

7. • The tunability of the tan delta seems very interesting, but the measurement seems to lack some context. The data transition from more elastic to viscous as r increases. Please see my

prior question as there needs to be a connection to the broader materials property attributes. The observations are likely related to rheology in polyelectrolyte complexes (reviews are available, by Larson for example) where a much wider polymer concentration can be studied and the effect of salt concentration. Few have examined the effect of concentration and molecular weight asymmetry.

Response: Thanks for the question. We agree with the reviewer that $\tan\delta$ values need to be connected to the broader materials property attributes. Besides, Larson *et al.* studied a broad range of polymer concentration and salt concentration in polyelectrolyte coacervates. Here we have studied the different charge ratio between the guest polyelectrolytes and host gels, and also the molecular weight of the guest chains in the complexation gel composites.

Now we have added the comment “ $\tan\delta$ values of the gel complexation composites can be tuned in the range of 0.06-1, which can match various soft tissues. For example, for bovine meniscus, $\tan\delta$ is in the range of 0.19-0.38 for different parts²⁷, and for bovine articular cartilage, $\tan\delta$ is around 0.13-0.2^{1,28}. For human lumbar, \tan is in the range of 0.42-0.58⁵. Besides, Larson *et al.* studied a broad range of polymer concentration and salt concentration in polyelectrolyte coacervates²⁹. Here we have studied the different charge ratio between the guest polyelectrolytes and host gels, and also the molecular weight dependence of the guest chains in the complexation gel composites.” in line 112-118. We have also cited the related references 1, 5, 27-29.

Introduction and Results and Discussion clarifications and Corrections

8. • The authors are making an equivalence between the DLS interpreted hydrodynamic mesh size and the static mesh size, this is not proven for polymer gels (201-207). The generalized Stokes Einstein relationship was found to apply to polyelectrolyte complex solutions, recently, such that the zero-shear solution viscosity, not the solvent viscosity, provides a more accurate interpretation of the cooperative diffusion coefficient for the hydrodynamic correlation length. In the present case, the solution viscosity may not be far from the solvent within a small factor depending on the concentration. Only one data point was brought up, but this was not shown for the different degrees of crosslinking or concentration in the present system.

Response: Thanks for the comments. We agree with the reviewer that for gel networks, the dynamical correlation length obtained from the generalized Stokes Einstein relationship measured by dynamic light scattering (DLS) is not equal to the mesh size of the gel matrix

obtained from Ornstein-Zernike equation measured by static light scattering. Previous studies have proven that the mesh size of the gel network can be obtained from Ornstein-Zernike equation measured by static light scattering (*Nat. Commun.*, 2018, 9 (1), 2248; *Phys. Rev. Lett.*, 2021, 126, 057802; *Macromolecules*, 2020, 53, 90–101). Here it is just by coincidence the value of dynamical correlation length obtained from DLS is close to the mesh size obtained from Ornstein-Zernike equation. The reason that the two values are very close is that the gel network has little heterogeneity and is quite homogeneous because the gel is highly charged with 80% charge density. However, our writing may confuse the readers about the two concepts of dynamical correlation length and mesh size obtained from static light scattering.

To clarify it, in line 228-230 in the revised version, we have changed “which is consistent with the static correlation length $\xi_{\text{static}} = (21.0 \pm 2.0)$ nm (Fig. 2a) measured by static light scattering. Therefore, both static and dynamic correlation length can be an estimation of the averaged mesh size of the gel network ($\xi_{\text{dynamic}} = \xi_{\text{static}} = 21$ nm).” into “Please note that although the measured values of the dynamic correlation length and the mesh size obtained from static light scattering are very close, the two concepts are not equal in terms of their physics.”

9. • The term “Entropy driven topological isovolumetric point” was coined, however, I don’t really see the data where no gel swelling occurs. I can see the boxes that show a trend between LPDA and SPDA and some points are at a swelling ratio of 1. A no swell or no shrink gel is very interesting, it would be more convincing of a full trend with more than two molecular weights were shown. Figure 6a does not show the MPDA (middle molecular weight data).

Response: Thanks for the comment. The figure for MPDA swelling ratio is in the Supporting Information of Fig. S3(d). We fully agree with the reviewer that the concept “isovolumetric point” we proposed here needs a more detailed and systematic study. In Fig. 6(a), only a few data points show the swelling ratio $\Phi/\Phi_0 = 1$ due to several competing physical parameters. For example, osmotic pressure and Donnan equilibrium can lead to swelling, while complexation between the guest chains and host gel can lead to shrinking. Therefore, this part is not the main discovery of this manuscript, and we wrote it only in the Outlook session (in line 566) in the revised version. Now we have also added the comment “A systematic study of the proposed “Entropy-driven Topologically Isovolumetric Point” is relegated to our future work.” in line 617-619 in the revised version.

10. • Line 372 the authors refer to a close to stoichiometric complexation at an r value $r=0.60$. Please explain or clarify as by the author's own definition of "r" is the ratio of the charges determined by the monomer charge and concentrations.

Response: Thanks for the comment. Please see the previous answer for question 5. The molar charge ratio "r" used in the work is an **apparent molar charge ratio**, which is calculated as the ratio of the positively charged monomers of PDA over the negatively charged monomers of the gel matrix in the pregel solutions in feed before the gelation chemical reaction starts. Because as the reviewer suggested, after gelation chemical reaction and dialysis, the true molar charge ratio values depend on several factors, such as the chemical conversion (Fig. 3b), dialyzed out, rejection of the free polymer from the network, degree of ionization of PDA, *etc.* These factors are very difficult to control so that the true molar charge ratio is very hard to quantify.

In Line 396-400 in the revised version, we refer to a close to stoichiometric complexation by considering both the monomer chemical conversion (Fig. 3b) and the apparent molar charge ratio in feed to roughly estimate the actual molar charge ratio, which is still a qualitative result. Because other factors such as rejection of the free polymer from the network cannot be quantified precisely.

To make it clear, now we have changed the statement "Based on the Ac monomer chemical conversion (Fig. 3b), for $r=0.6$ the actual molar charge ratio of LPDA/ gel matrix is 0.79, which is" into "For the apparent charge ratio $r=0.6$, we can roughly estimate that the actual molar charge ratio of LPDA/ gel matrix is 0.79 if the monomer chemical conversion is considered. It is" in line 396-398 in the revised version.

11. • What is the meaning of the dotted circles in Figure 5a and 5b?

Response: The dotted circles highlight the different trends for the samples with LPDA (Fig. 5a) and SPDA (Fig.5b) respectively. G' values of the gel composites with SPDA swelled by SPDA solutions are all lower than G' of their as-prepared states (green dashed oval in Fig. 5b), but G' of the gel composites with LPDA swelled by LPDA solutions are slightly higher than G' of their as-prepared states (green dashed oval in Fig. 5a). This statement has been mentioned in line 472-474 in the revised version.

12. • What are the authors trying to explain with Figure 5c and 5d?

Response: Thanks for the question. For physical gels with ionic bonds, they can be well described by the sticky Rouse motion of associating polymer strands derived from ionic bonds (sticker) with $G' \sim \omega^n$, $G'' \sim \omega^m$ ($n=m=0.5$). While for chemical gels, G' is independent of frequency ($n=0$). Besides, Larger ($m-n$) values indicate the sample is more elastic in nature, and smaller ($m-n$) values indicate it is physical gels with ionic bonds. While our gel complexation composite has both physically ionic bonds and chemical bonds. **So in Figure 5c and 5d, the trends of n and m exponents can reflect different dominant structures and physical interactions such as chemical gels, physical gels with ionic bonds, and “repulsive gel composites”.** For example, for LPDA gel composites in the as-prepared state, n exponents increase with r indicating that the system transfers from chemical gels to physical gels with ionic bonds with increasing r (Fig. 5c). While for the SPDA gel composites in the as-prepared state, n values first increase and then decrease, and n even decrease to 0 at $r=2$ for the as prepared state (Fig.5d), indicating they become “repulsive gel composite” dominating by the repulsive interaction. The different behaviors of samples with LPDA and SPDA are due to their different topologically correlated structures between the guest chains and host gels arising from the size ratio of the guest PDA chain size over the mesh size of the host gel.

Overall, because of various topologically correlated structures of the complexation gel composites, for gel composites with LPDA, as r increases, they will have the chemical gels to physical gels with ionic associations to topologically quasi-interpenetrating network structural transition. While for gel composites with SPDA, as r increases, they will have the chemical gels to physical gels with ionic bonds to “repulsive gel composites” structural transition. The detailed explanations about n , m values in Fig. 5c and 5d are in line 501-524, and line 537-541 in the revised version.

13. • Authors attribute an “enthalpy effect” for complexation on line 355 to 358. Provide a citation as to who determined this to be true as it is not universally accepted.

Response: Thanks for the excellent comment and good catch! Although we used “enthalpy effect” to try to describe the electrostatic attraction energy between the polycation guest chains and negatively charged host gel, actually for the coacervate systems the counterion release, which is an entropy effect, is the main driving force for the coacervation. It is indeed not fair to attribute an “enthalpy effect” for complexation. Thanks for the reviewer’s good catch!

Now we have changed the statement “Please note that the enthalpy effect, which arises from the electrostatic complexation between the LPDA and the unoccupied charged sites on the gel matrix, is the driving force to make the LPDA chains diffuse inside the gel composites.” into “Please note that the electrostatic complexation between the LPDA and the unoccupied charged sites on the gel matrix is the driving force to make the LPDA chains diffuse inside the gel composites.” in line 380-382 in the revised version.

Response to Referee #2

We thank the reviewer's warm comment that the manuscript is highly enjoyable to read. We also thank the reviewer for appreciating the clear writing and appropriate level of relatively complex theory. We are also grateful to the reviewer for several excellent comments. Those comments and suggestions have improved the quality of the manuscript significantly. We have addressed all points raised by the reviewer as detailed below and also in the revised manuscript. All the changes are highlighted in the revised manuscript. The comments of the reviewer are in black and our response is in blue.

This manuscript presents a new class of hydrogel materials wherein viscoelasticity and swelling can be independently controlled. The materials have potential biomaterial applications as their range of shear moduli are similar to several tissues of importance in regenerative medicine. The experimental design, methodology, analysis, interpretation and presentation are outstanding. Concerns primarily relate to weak support for the materials in biomedical applications and some sections of the text being challenging to follow.

Response: We are grateful for the warm endorsement on novelty of our findings and for the nice suggestion. We also thank the reviewer for appreciating the experimental design, methodology, analysis, and data interpretation. We agree that the paper must be reasonably easy to follow for even non-experts in this field, especially some sections of the text being hard to follow for the readers. Now we have made changes throughout the manuscript.

We also agree with the reviewer that more supports are need for biomaterials and biomedical applications, but in this paper we mainly focus on the physical mechanism by using a physical model system, which is composed of guest charged macromolecules complexed with the oppositely charged host gel. For example, how the viscoelasticity and swelling can be independently controlled, how to precisely control the relationship between the complexation structure and viscoelastic properties by utilizing a newly discovered physical mechanism of entropy-driven topologically frustrated dynamical state. **The physical mechanism discovered in this work is ubiquitous and can be applied to all kinds of biomaterials and natural polymers as long as they can fit our physical model system which is composed of guest charged macromolecules complexed with the oppositely charged host gel with designed size ratio of the macromolecular chain size over the gel mesh size.** One example of real biosystem which can fit our physical model is a complexation gel composite which is

composed of positively charged poly-lysine (or other positively charged poly(amino acids)) as guest chains and negatively charged hyaluronic acid gel matrix as host gel with designed size ratio of poly-lysine chain size over the hyaluronic acid gel mesh size. We will be very happy and grateful if our physical model system can be applied to all kinds of real biomaterials and biomedical applications in the future by the experts in biomaterials and biomedical field.

Biomedical applications: The polymers selected have potential in biomaterials applications such as cartilage repair in articulating joints and the spine. As mentioned in the manuscript, the hydrogels' shear moduli range is relevant to that of cartilage and vitreous fluid. Cartilage repair has been a particularly important area in regenerative medicine, with top leaders in the biomaterials field (Anseth, Burdick, Khademhosseini to name a few) investing major research focus over decades on hydrogels for cartilage repair. The reported hydrogels have potential to significantly impact this field. However, several concerns are the following:

Response: Thanks for the reviewer's appreciation that the complexation gel composites reported in this work have potential to significantly impact in the cartilage repair biomaterials field. The answers to the reviewer's questions and concerns are detailed below.

1. An approach to repairing cartilage is to permanently patch an injury or replace damaged tissue with a non-degradable material. It is presumed that the reported hydrogels are not degradable under physiologic conditions because polyacrylamide is well known to be non-degradable and a brief literature search suggests that poly(diallyldimethylammonium chloride) (PDA) may be non-degradable as well. It would be helpful for the authors to briefly address this topic.

Response: That's a good point! Both polyacrylamide and PDA have pure carbon backbones, and the degradation of this type of polymers usually required high temperature, radiation, or high shear / elongational forces, *etc.*, and those conditions have little chance to occur in vivo (Francis. S., Varshney, L., Sabharwal, S. Eur. Polym. J. 43, 2525 (2007); Xiong, B. et al. npj Clean Water 1, 17 (2018)). Therefore, both polyacrylamide and PDA are relatively stable and non-degradable in vivo.

Overall, this paper mainly focused on discovering a ubiquitous physical mechanism by using a physical model system, and the ubiquitous physical mechanism discovered in this work can

be applied to all kinds of biomaterials and natural polymers as long as they can fit our physical model system which is composed of guest charged macromolecules complexed with the oppositely charged host gel with designed size ratio of the macromolecular chain size over the gel mesh size. There are wide variety of charged polymer that can be degradable and non-degradable for different purposes, for example degradable polycation poly-L-lysine bromide as drug carrier (Patil and Kandasubramanian. *Eur. Polym. J.* **146**, 110248 (2021)), and non-degradable polyanion poly(2-acrylamido-2-methyl propanesulfonic acid) was used as artificial supporting tissues. (Yasuda and Gong *et al. Adv. Mater.* **28**, 6740–6745 (2016)) The applications of our physical model to all kinds of real biomaterials and biomedical applications are relegated to the future work by the experts in biomaterials and biomedical field.

Now we have added the comment “The applications of our physical model and mechanism need specific requirement for the materials. For example, an approach to repairing cartilage is to permanently patch an injury or replace damaged tissue with materials which are a non-degradable and non-toxic, and polyacrylamide and PDA used here are non-degradable and non-cytotoxic in vivo⁷¹⁻⁷⁴.” in line 549-553 in the revised version. And we have also cited the related references 73-74: Xiong, B. *et al. npj Clean Water.* **1**, 17 (2018); Francis, S., Varshney, L. & Sabharwal, S. *Eur. Polym. J.* **43**, 2525-2531 (2007).

2. Along similar lines, polyacrylamide (when washed to remove any residual acrylamide monomers) is not cytotoxic and a brief literature search indicates that PDA is potentially antimicrobial but not toxic to mammalian cells. Please comment on this topic as well.

Response: Applications of polyacrylamide and PDA into biomaterials are well established and they have been proved to be not cytotoxic in vitro and in vivo (Mukherjee *et al. ACS Appl. Nano Mater.* **3**, 5826–5837 (2020); Balzani & Guck *et al. Sci. Rep.* **9**, 17031 (2019); Yu *et al. Nano Energy* **109**, 108324 (2023)). Similar to the answer to the previous question, these two polymers are just a physical model, more biocompatible polymer material can be selected, depending on the needs of the specific applications.

Now we have added the comment “The applications of our physical model and mechanism need specific requirement for the materials. For example, an approach to repairing cartilage is to permanently patch an injury or replace damaged tissue with materials which are a non-degradable and non-toxic, and polyacrylamide and PDA used here are non-degradable and non-cytotoxic in vivo⁷¹⁻⁷⁴.” in line 549-553 in the revised version. And we have also cited the related

references 71-72: Saini, B., Singh, R. R., Nayak, D. & Mukherjee, T. K. *ACS Appl. Nano Mater.* 3, 5826-5837 (2020); Träber, N. et al. *Sci. Rep.* 9, 17031 (2019).

3. Hydrogels exist that span the same range of shear modulus. For example, this reviewer has unpublished data that polyacrylamide gels of varying bis:acrylamide span a shear modulus range of 10-10,000 Pa. While swelling ratios were not measured, massive changes in swelling were visually observed, with swelling inversely proportional to shear modulus. A brief discussion supporting the need for a class of hydrogels with these ranges of tunable shear moduli as well as swelling (from shrinking, to isovolumetric, to highly swelling) in biomedical applications would significantly strengthen the rationale for this manuscript.

Response: Thanks for the excellent suggestion! We have added a new plot in Fig. 1(g) to make a comparison of our complexation gel composites, synthetic hydrogels, and biomaterials. It clearly shows that our complexation gel composites mainly fill the gap in the ultrasoft-soft regime. Our proposed hydrogel system basically covered from mucus, collagen-proteoglycan mixtures, vitreous humor, articular cartilage, and nucleus pulposus *etc.*. It is clear that hydrogels with moduli in this range ($1\sim 10^5$ Pa) hold great potential for biomedical applications.

In terms of swelling behavior of hydrogels for biomaterials and biomedical applications, based on a review article (Feng and Wang, *Adv. Sci.* 10, 2303326 (2023), <https://doi.org/10.1002/advs.202303326>), hydrogel can be classified into 3 types as High swelling(>150%), Non-swelling(0-150%) and Shrinkable(<0). While High swelling hydrogels are for tissue engineering and drug delivery due to their high cell recruitment and migration and ability for molecules diffusion and release. Non-swelling hydrogels are favored for tissue engineering and bioelectronic due to their mechanical properties and stability, but lacking ability to load and release drug. Shrinkable hydrogels show great potential for scaffold as their stiffness and promotion for cell attachment. In our work, swelling ratio can be tuned from highly shrinking ($\Phi/\Phi_0=4.16$), to isovolumetric ($\Phi/\Phi_0=1$), to highly swelling ($\Phi/\Phi_0=0.08$) in Fig.6a. More importantly, the viscoelasticity and swelling can be independently controlled in our complexation gel composites, so that they can fit in various biomedical applications.

Now we have added this plot in Fig. 1(g) and have added the comments “A comparison of the complexation gel composites in this work with synthetic hydrogels and soft tissues has been made. It clearly shows that our complexation gel composites mainly fill the gap in the ultrasoft-soft regime.” in line 53-56 in the revised version. We have also cited the references 1-5, 28, 39-

44 in the caption of Fig. 1(g).

We have also added the comment about swelling in biomedical applications as “In this model system, swelling ratio can be tuned from highly shrinking to isovolumetric to highly swelling. More importantly, the viscoelasticity and swelling can be independently controlled in our complexation gel composites, so that they can fit in various biomedical applications. For example, high swelling hydrogels are for tissue engineering and drug delivery due to their high cell recruitment and migration and ability for molecules diffusion and release. Non-swelling hydrogels are favored for tissue engineering and bioelectronic due to their mechanical properties and stability, but lacking ability to load and release drug. Shrinkable hydrogels show great potential for scaffold as their stiffness and promotion for cell attachment⁷⁵.” in line 607-615 in the revised version. And we have also cited the reference 75. Feng, W. & Wang, Z. *Adv. Sci.* 10, 2303326 (2023).

4. For biomedical applications, physiologic conditions are the most relevant for characterization. Typically, materials are evaluated in phosphate buffered saline (PBS). Is there any way to relate the findings of this work to a physiologic ionic strength? This is of particular concern for the gels equilibrated in water, as the shear moduli of these gels range 100-10,000 Pa, yet samples “as prepared” (not equilibrated in a solution following polymerization) and samples equilibrated in PDA solutions only range 10-1000 Pa.

Response: Thanks for the good question. The physiologic ionic strength is 0.154M and PBS buffer has similar ionic strength. For our complexation gel composites, when PDA complex with poly (acrylamide-co-sodium acrylate) gel with 80% charge density, counterions (NaCl) will be released and thus the ionic strength of the system will increase. When the molar charge ratio r changes from 0 to 1, the ionic strength of the system can cover the range of 0-0.4M due to counterions (NaCl) release. So our gel system has covered the physiologic ionic strength.

Secondly, the advantage of our current system is that the moduli and viscoelasticity can change with the environment’s ionic strength. For example, in vitro at low ionic strength the modulus of the materials is high, but when it is transferred in vivo under physiologic conditions, the materials will change to be softer with lower elastic modulus. Because in our gel system, in addition to the chemical bonds, physically ionic bonds due to complexation between the guest chains and host gel play a major role in determining the viscoelasticity and moduli. The strength of the ionic bonds is relatively high at low ionic strength when equilibrated in water.

While at high ionic strength, the electrostatic interaction is screened and the strength of the ionic bonds is too weak to contribute to the moduli, thus the moduli is relatively low.

Thirdly, the moduli of non-charged hydrogels such as polyacrylamide gels can be independent of ionic strength, so that they can have high moduli at both high ionic strength (such as physiologic conditions) and low ionic strength, but their moduli lack the tunability by changing ionic strength, so it really depends on the specific needs for specific applications. In our current system, the gel matrix is highly charged (with 80% charge density), we can systematically tune the charge density of the gel matrix to change the moduli dependency on the ionic strength. When the charge density decreases to 0, it will become non-charged hydrogels, thus their moduli will become stable and independent of ionic strength. The gel charge density investigations to tune the moduli dependency on the ionic strength, and studies for a similar gel system under physiologic conditions are relegated to the future work.

Now we have added the comment “For the as-prepared gels, the ionic strength of the system is in the range of 0-0.4M due to counterions release, depending on the molar charge ratio r .” in line 449-450 in the revised version. And also added the comment “The gel charge density investigations to tune the moduli dependency on the ionic strength, and studies for a similar gel system under physiologic conditions are relegated to the future work.” in line 547-549 in the revised version.

Other concerns: The manuscript is highly enjoyable to read because of its clear writing and appropriate level of relatively complex theory (motivation, explanation and use in analysis). However, there are some areas that should be addressed in order to yet improve the ease of reading of the work.

Response: We thank the reviewer’s warm comment that the manuscript is highly enjoyable to read. We also thank the reviewer for appreciating the clear writing and appropriate level of relatively complex theory. We agree that we need to improve the ease of reading of the work for the readers, especially some sections of the text being hard to follow for the readers. Now we have made changes throughout the manuscript as detailed below.

5. The paragraphs are very long (and will appear yet longer when formatted for Nature Communications), making them hard to follow. It is highly recommended to break up

paragraphs -- in areas such as,

- Line 119, where the introduction transitions from background to description of the work herein

- Line 493, where results of m and n transition to $\tan(\delta)$

These are but two examples that were of note to this reviewer, but it is recommended that consideration should be given to breaking up most paragraphs throughout.

Response: Thanks for the good suggestion. The long paragraphs have been broken up so that they are easier for readers to follow now.

6. Extremely minor issue: line 42 states “rad/s⁵” (where 5 is a citation) but at first it was read to be a typo suggesting rad/s². Not sure what can be done here, but wanted to mention it because rad/s² would be confusing.

Response: Thanks for the careful reading. We have changed “And for the medium soft case is the nucleus pulposus, which has a frequency-sensitive viscoelastic behavior, has an elastic modulus G’ in the range of 3–7 kPa tested over the range of angular frequencies 1-100 rad/s⁵.” into “And for the medium soft case is the nucleus pulposus, which has a frequency-sensitive viscoelastic behavior⁵, has an elastic modulus G’ in the range of 3–7 kPa tested over the range of angular frequencies 1-100 rad/s.” in line 41-43 in the revised version.

7. Line 45: “modulate their moduli” is repetitive.

Response: Now we have changed “modulate” into “change”. Thanks!

8. In lines 158-159, LPDA, MPDA and SPDA are defined. On first reading, it was assumed that L stands for low, M stands for medium and S was unclear. When noticing the MW ranges for each acronym, it was then assumed that L stands for “large” and M stands for “medium” and S stands for “small.” This notation is fine, but because ranges are typically ordered as low→high, it is suggested that the acronyms are listed as such and defined.

Response: We have changed the list order from low to high in line 179-180 in the revised version. Thanks for the careful reading!

9. Lines 259-262 contain essential information towards interpretation of the results. By breaking up the paragraph or another means, it would be helpful if these sentences were more obvious.

Response: Thanks for the good suggestion. Now we have broken up the paragraph to make the essential information more obvious.

10. Line 362: Is “synergistic” meant instead of “synergetic”? They might mean the same thing, but synergistic is more commonly used.

Response: Thanks for the careful reading. Now we have changed “synergetic” into “synergistic”.

12. For Figure 4a-d, it would be helpful to point out in the caption or text that the y-axis ranges are important to note.

Response: Thanks for the suggestion! We have added “The y-axis ranges are important to note.” in the figure caption.

13. Line 399 is confusing. Are the values reported in Figure 4d actual/2.5? Please clarify this normalization and substitute “shifted” for a more specific term.

Response: Yes, the values reported in Figure 4d for PDA swelling (red curves) are the actual values divided by 2.5 for clear vision. Otherwise, the data for As-prepared (black curves) and for PDA swelling (red curves) in Figure 4d will partially overlap and mix together so that they cannot be clearly displayed. Now we have changed “shifted” into “divided”. Thanks!

14. All figures: The placement of (a), (b), etc. varies throughout. It would be helpful if they were consistently located in the same corner of each panel throughout.

Response: Thanks for the suggestion! Now we have placed all the (a), (b), *etc.* on the upper left corner of the figures.

15. Line 487: It is suggested that “besides” be replaced with “in contrast, “however” or “yet”

Response: Thanks for the suggestion. Now we have changed “Besides” into “However”.

16. Figures 5 and 6: Please write out or define “s” and “As-p” in the symbol legend or caption.

Response: Thanks for the careful reading. “s” is shorten for swelling and “As-p” is shortened for As-prepared in all the figures. Now we have defined “s” and “As-p” in the figure caption of Figures 5 and 6.

17. Figure 6b: What do I, II and III denote? Also, orange letters on the green background are hard to distinguish.

Response: Thanks for the reviewer’s careful reading. I, II and III denote three possible dependent pathways for the swelling ratio Φ/Φ_0 as a function of guest chain size R_g . All of them would pass through the point of $\Phi/\Phi_0 = 1$, which is defined as an isovolumetric point. Now we have added the comments “I, II and III denote three possible dependent pathways for the swelling ratio Φ/Φ_0 as a function of guest chain size R_g ” in the figure caption of Fig. 6b. This is also mentioned in line 606-607 in the revised version.

The orange color has also been changed. Thanks!

Reviewers' Comments:

Reviewer #1:

Remarks to the Author:

The authors have understood my review and made changes to the manuscript including the new Figure 1, polymer characterization analysis and reporting of values, and current working hypothesis of entropy-driven complexation to name a few. The work is original with a balance of measurement and analysis that should inspire a broad readership in the soft matter community. Others with more applied and clinical interests will benefit by the trends and can tune specific polymers or biopolymers guided by fundamentals. I recommend accepting in Nature Communications.

Reviewer #2:

Remarks to the Author:

Overall, the authors adequately addressed Reviewer 2 concerns except for comment 13 about scaling a data set in Figure 4d. It is appreciated that some data is difficult to depict clearly because of overlapping features. Perhaps a second y-axis for the scaled data set would make the authors' intention clearer.

Response to Referee

The reviewers' comments are as follows:

Reviewer #1 (Remarks to the Author):

The authors have understood my review and made changes to the manuscript including the new Figure 1, polymer characterization analysis and reporting of values, and current working hypothesis of entropy-driven complexation to name a few. The work is original with a balance of measurement and analysis that should inspire a broad readership in the soft matter community. Others with more applied and clinical interests will benefit by the trends and can tune specific polymers or biopolymers guided by fundamentals. I recommend accepting in Nature Communications.

Response: Thanks for the reviewer's great efforts to revise and improve the quality of the manuscript. With the reviewer's help, we believe now the manuscript is much clearer and attractive to a broad range of readers with different backgrounds such as biomaterials, physics, tissue engineering, *etc.*
Thanks!

Reviewer #2 (Remarks to the Author):

Overall, the authors adequately addressed Reviewer 2 concerns except for comment 13 about scaling a data set in Figure 4d. It is appreciated that some data is difficult to depict clearly because of overlapping features. Perhaps a second y-axis for the scaled data set would make the authors' intention clearer.

Response: Thanks for the reviewer's great efforts to revise and improve the quality of the manuscript. Now we have added a second y-axis for Figure 4d, as the reviewer suggested. Please see the revised Figure 4d in the revised manuscript. With the reviewer's help, we believe now the manuscript is much clearer and attractive to a broad range of readers with different backgrounds such as biomaterials, physics, tissue engineering, *etc.* Thanks!